# Memory-Augmented Personalized Retrieval for Long-Context Egocentric Video

## Abstract

Recent advances in AI and wearable devices, such as augmented-reality glasses, have made it possible to augment human memory by retrieving personal experiences in response to natural language queries. However, existing egocentric video datasets fall short in supporting the personalization and long-context reasoning required for episodic memory retrieval. To address these limitations, we introduce EgoMemory, a benchmark derived from Ego4D, enriched with 165,795 user-specific object annotations over 245 videos from 45 participants, yielding 639 distinct, human-curated, and evaluated queries for rich and individualized episodic memory retrieval. Leveraging this resource, we present `EgoRetriever`, a novel, training-free retrieval framework that combines Multimodal Large Language Models with reflective Chain-of-Thought prompting. Our approach enables interpretive inference of user intent and generates detailed target video descriptions by leveraging contextualized personal memory for video retrieval. Extensive experiments on EgoMemory, EgoCVR, and EgoLifeQA benchmarks demonstrate that `EgoRetriever` consistently and substantially outperforms state-of-the-art baselines, highlighting its strong generalizability and practical potential for personalized, long-context egocentric video retrieval.

## 1 Introduction

The integration of AI into wearable technologies (*e.g.*, glasses), suggests a future where human memory is augmented through continuous experience capture and retrieval. This notion closely resembles Vannevar Bush's "Memex", proposed in 1945 as a conceptual system for amplifying cognition through personalized, associative information access Bush et al. (1945), which defines personalization as grounding retrieval in the specific objects a user has seen, experienced, or interacted with in humans' daily life. Recent advances in wearable devices and large language models (LLMs) bring this long-standing vision within reach.

Central to realizing this vision is the task of episodic memory retrieval Grauman et al. (2022), which aims to extract relevant visual episodes from a user's egocentric video archives based on natural language queries. Distinct from traditional text-to-video retrieval, this task uniquely emphasizes *personalization*: (i) data are continuously recorded from the user's viewpoint; (ii) most queries explicitly reference personal objects (*e.g.*, our empirical analysis in Section 3.2 indicates that **88.4%** of queries in the Ego4D dataset Grauman et al. (2022) exhibit such explicit referencing); (iii) user queries frequently involve specific objects or actions in history memory (*e.g.,* "what is the location I play with my dog in last month?"), necessitating models capable of long-context video understanding (*i.e.,* spanning months). This naturally motivates a dynamic personal memory bank that accumulates recurring objects, habits, and social interactions. Since personal interactions repeatedly involve the same items, object frequency is a strong cue of personalized relevance in both intuition and recent works Lee et al. (2012); Lee & Grauman (2015); Grauman et al. (2022); Yang et al. (2025). Enriching user queries with contextualized personal information thus holds significant potential for improving long-context memory retrieval accuracy. Nonetheless, current episodic memory retrieval tasks predominantly concentrate on single-video scenarios Grauman et al. (2022); Hummel et al. (2024) or relatively short-term contexts (*i.e.,* within one week) Yang et al. (2025), neglecting the personalized, long-context nature intrinsic to episodic memory retrieval.

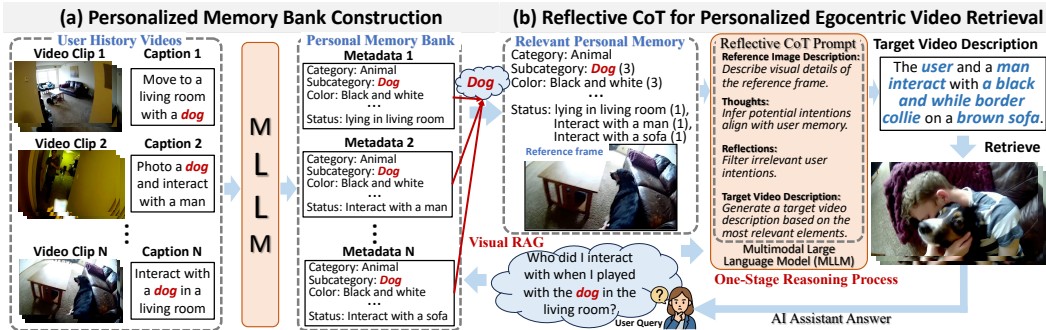

Figure 1: Overview of our approach for personalized egocentric video retrieval, comprising two progressive modules: (a) offline construct a personalized memory bank from each user's historical videos; (b) online retrieval by query-relevant personal memory to guide intention understanding.

To address this limitation, our study focuses explicitly on long-context personal egocentric memory retrieval. Given the absence of explicit annotations for personally relevant objects in existing egocentric video datasets Singh et al. (2016); Grauman et al. (2022); Hummel et al. (2024); Yang et al. (2025), we propose an annotation pipeline leveraging the advanced reasoning capabilities of MLLMs. Specifically, we apply this pipeline to annotate 245 videos from 45 unique participants in the Ego4D dataset Grauman et al. (2022), resulting in 165,795 user-specific object annotations to constitute a comprehensive personal memory bank. All candidate queries are further filtered for personalization and long-context via an MLLM-assisted procedure with final human verification (∼91.6% personal, details in Section 4.1). Based on this, we introduce the **EgoMemory** benchmark, designed explicitly for learning personalized information from users' historical videos to facilitate long-context episodic video retrieval (details in Section 3.2). Figure 1(a) exemplifies this by demonstrating how attributes of a user's personal item (a "dog") can be systematically extracted from past video clips and corresponding captions via MLLMs. During inference, the user's query is analyzed to identify relevant personal objects (*e.g.,* "dog"), after which pertinent memories are retrieved from the personal memory bank to enrich the query and improve subsequent video retrieval precision.

Utilizing the relevant personal memory, we propose `EgoRetriever`, a novel, training-free framework tailored explicitly for long-context episodic video retrieval. As shown in Figure 1(b), `EgoRetriever` combines Multimodal Large Language Models (MLLMs) with a reflective Chain-of-Thought (CoT) prompting strategy to infer nuanced user intentions and generate detailed textual descriptions of target video clips. This approach enhances retrieval accuracy by incorporating fine-grained details, such as the dog's color and contextual elements like "sofa" and "interaction with a man", drawn from personal memory.

To summarize, the main contributions are: (1) We introduce a memory-augmented framework for the personalized long-context egocentric video retrieval task and present the **EgoMemory** benchmark, which features individualized memory banks constructed from extensive user-specific object annotations in Ego4D Grauman et al. (2022).(2) We propose **`EgoRetriever`**, a training-free retrieval framework that combines Multimodal Large Language Models (MLLMs) with reflective Chain-of-Thought (CoT) prompting to interpret user queries by leveraging personal memory and generate detailed descriptions for video retrieval. (3) Extensive experiments on both the EgoMemory and EgoCVR Hummel et al. (2024) and EgoLifeQA Yang et al. (2025) benchmarks demonstrate that `EgoRetriever` consistently and significantly outperforms existing baselines, highlighting its strong generalizability and its potential for real-world deployment in egocentric video retrieval.

## 2 RELATED WORK

**Egocentric Datasets and Benchmarks.** Early egocentric studies used ADL Pan et al. (2022), CharadesEgo Sigurdsson et al. (2018), and EGTEA Gaze+ Li et al. (2018), but these were limited in scale and diversity. Larger datasets (*i.e.,* EPIC-KITCHENS Damen et al. (2020) and Ego4D Grauman et al. (2022)) broadened the field and enabled many tasks. Specialized corpora, including EgoProceL Bansal et al. (2022), IndustReal Schoonbeek et al. (2024), HoloAssist Wang et al. (2023a), EgoExo4D Grauman et al. (2024), and EgoExoLearn Huang et al. (2024), target procedural and

multi-view understanding. Recent benchmarks such as EgoSchema Mangalam et al. (2023) and EgoPlan-Bench Li et al. (2024) emphasize temporal reasoning and planning, while EgoMemoria Ye et al. (2025) and EgoLife Yang et al. (2025) provide week-long, multi-participant data for studying longer-term behavior. However, these benchmarks generally overlook the fine-grained, person-specific variability needed for long-context *personalized* retrieval. Our EgoMemory addresses this gap by offering the first benchmark for personalized egocentric video retrieval, explicitly capturing inter-individual daily variability and enabling person-centric memory augmentation.

**Composed Image and Video Retrieval.** Composed image retrieval (CIR) retrieves images that are semantically edited by textual prompts Vo et al. (2019); Baldrati et al. (2022). Zero-shot CIR methods Saito et al. (2023); Baldrati et al. (2023); Tang et al. (2024d); Gu et al. (2024); Karthik et al. (2024); Tang et al. (2024c); Suo et al. (2024); Du et al. (2024); Tang et al. (2024b; 2025) use multimodal encoders such as CLIP Radford et al. (2021) to reduce annotation needs, yet often struggle with implicit human intent. Recent training free approaches (e.g., CIReVL Karthik et al. (2024) and OSrCIR Tang et al. (2024a)) leverage large language models to infer intent and improve compositional reasoning without supervision. Extending to video, composed video retrieval addresses temporal complexity. EgoCVR Hummel et al. (2024) supports fine-grained egocentric queries with a two-stage caption fusion pipeline. Despite progress, current frameworks are still under a model dynamic context and personal relevance in real egocentric scenarios. We introduce a training-free, one-stage retrieval framework that grounds user queries in a dynamic personal memory bank and produces fine-grained, context-aware video descriptions. This design achieves state-of-the-art performance on EgoMemory and advances personal memory retrieval.

**Memory Augmented Long Context Retrieval.** Retrieval Augmented Generation (RAG) frameworks show that coupling large language models with external memory extends reasoning over long contexts Lewis et al. (2020); Jiang et al. (2023); Shi et al. (2023); Ram et al. (2023); Izacard et al. (2022). Graph augmented retrieval improves multi-hop reasoning by using structured knowledge graphs built from sources such as Wikipedia and document-level entities for re-ranking and contextual linking Ding et al. (2019); Zhu et al. (2021); Nie et al. (2019); Das et al. (2019); Asai et al. (2020); Li et al. (2021). For example, HippoRAG Gutiérrez et al. (2024) employs neurobiologically inspired knowledge graphs for advanced reasoning. Lifelogging systems Rossetto et al. (2020); Nguyen et al. (2021) organize personal data with multimodal knowledge graphs but are constrained by static schemas and limited flexibility for dynamic, user-driven interpretation. In contrast, EgoMemory builds a personalized memory bank directly from egocentric video, and `EgoRetriever` uses a reflective chain-of-thought prompting within a training-free architecture to enable dynamic, user-specific reasoning and to produce detailed video descriptions. This yields significant gains over prior baselines for long context personal retrieval. Additional related work on multimodal *chain of thought* is provided in the Appendix A.13.

## 3 METHODOLOGY

In this section, we first formalize the proposed long-context video retrieval task with personalized memory augmentation. We then provide detailed descriptions of the two core components: the construction of personalized memory banks and the design of the `EgoRetriever` framework.

### 3.1 PRELIMINARY

We adopt a continuous collection setting where each user records egocentric videos $\mathcal{V} = \{V^{(1)}, \ldots, V^{(N)}\}$ over time and contexts. Recordings of varying duration are segmented in postprocessing into semantically coherent clips, yielding the candidate set $\mathcal{C} = \{C_1, \ldots, C_M\}$ for retrieval.

Personalization requires historical evidence. For each user, we build a memory bank $\mathcal{M}$ with a pretrained MLLM $\Psi_M$ (Section 3.2). The retrieval task is: given a natural language query $Q$, retrieve the most relevant clip $C^* \in \mathcal{C}$. We first parse $Q$ and query $\mathcal{M}$ to obtain personal object metadata $\mathcal{M}_q$ and a lightweight visual anchor $I_r$. To avoid conflating scene exposure with personalization, retrieval into $\mathcal{M}_q$ uses semantic match together with cross-video recurrence and possessive cues.

With $(Q, \mathcal{M}_q, I_r)$, `EgoRetriever` uses an MLLM with reflective Chain of Thought prompting to produce a focused target description $T_t$. A video language retriever (*e.g.*, EgoVLPv2 Pramanick et al. (2023)) embeds $T_t$ via a text encoder $\Psi_T$ and each candidate $C_i$ via a video encoder $\Psi_V$. Long

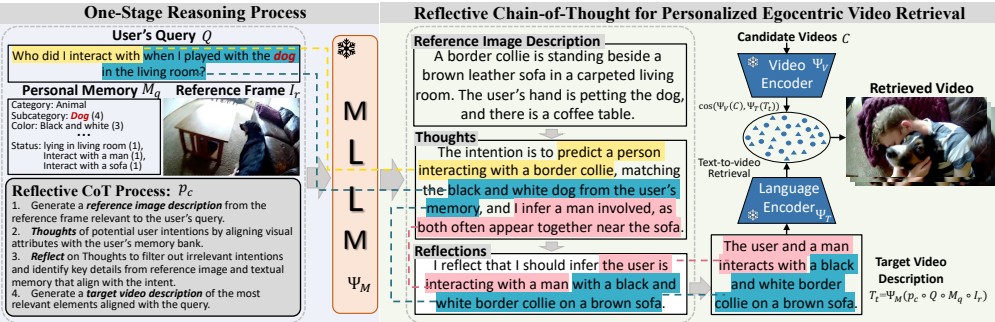

Figure 2: Overview of `EgoRetriever`. An MLLM processes textual personal memory data $M_q$, the reference frame $I_r$ and the user's query $Q$ to generate the desired target video description $T_t$ by reflective CoT. A vision-language model is then adopted to perform text-to-video retrieval. Texts with different colors show the reasoning traces of each user's intention.

context retrieval is performed by ranking candidates with cosine similarity $\cos(\Psi_V(C_i), \Psi_T(T_t))$, returning the top-ranked clips.

## 3.2 PERSONALIZED MEMORY BANK

Central to personalized retrieval is incorporating long-term, user-specific context rather than incidental scene exposure. From an empirical analysis of Ego4D Grauman et al. (2022) queries using SpaCy Honnibal et al. (2020) and WordNet Miller (1995)), **88.4%** explicitly reference physical objects, often linked to the user (*e.g.,* "my bag", "our dog"). Following prior works Lee et al. (2012); Lee & Grauman (2015); Yang et al. (2025), we operationalize *personalization* as *personally experienced objects*: entities recurring across a user's videos or linked to the user. To avoid conflating environment priors with personalization, we record cross-video recurrence, within-user attribute consistency, and possessive cues, and down-weight one-off co-occurrences. Specifically, the *Personalized Memory Bank Construction* encodes user-specific object metadata from long-context egocentric video(e.g., attributes, recurrence statistics, and visual exemplars) into a structured memory $\mathcal{M}$. The *Visual Retrieval-Augmented Generation* then fuses the query with memory entries that are semantically matched (as shown in Figure 1) and supported by cross-video evidence, favoring user-specific contextual cues over incidental context and improving long-context personal video retrieval.

**Personalized Memory Bank Construction.** We construct the memory bank with a pretrained MLLM $\Psi_M$[1], integrating video and narration. In EgoMemory, Ego4D narrations and clips are processed by $\Psi_M$ to extract object attributes (major category, subcategory, texture, shape, color, brand; prompts in Figure 7) *and* weak personalization cues (first-person/possessive mentions, coreference) together with cross-video recurrence statistics. For example, a *dog* is encoded as *animal→dog* with color/state and recurrence within the same user. Aggregating these profiles across clips yields a structured, user-specific memory $\mathcal{M}$ with entries $\mathbf{m}_i = \Psi_M(C_i)$. Unlike unstructured transcript baselines (*e.g.,* EgoLifeQA Yang et al. (2025)), our representation explicitly models user-specific contextual cues via recurrence and linguistic cues, helping reduce the weight of incidental scene exposures. Practically, since egocentric capture includes idle periods (e.g., sleep), construction can run asynchronously with minimal impact on interactive use.

**Visual Retrieval-Augmented Generation.** The memory bank serves as a semantic repository of personal experiences. Given a query $Q$, **Visual RAG** proceeds: (1) extract the object subcategory and retrieve $\mathcal{M}_q \subset \mathcal{M}$ using semantic match *and* recurrence/possessive cues (to reduce environment priors); (2) summarize attribute distributions in $\mathcal{M}_q$ into compact textual context; (3) pick a reference frame $I_r$ by selecting a reference video $V_r$ (centroid of the retrieved set) and taking its middle frame. $I_r$ is a lightweight visual anchor providing historical context and need not appear in the target clip $C^*$. Reflective CoT prompting in `EgoRetriever` then reasons over plausible evolutions (object/state/location), producing a focused description $T_t$ that prioritizes user-linked cues over incidental context. Thus, the enriched tuple $(Q, \mathcal{M}_q, I_r)$ improves personalized long-context video retrieval. For details, please refer to Appendix A.1.

---

[1]Efficiency aspects (*e.g.,* incremental extraction/updating) are orthogonal and deferred to future work.

### 3.3 REFLECTIVE CHAIN-OF-THOUGHT FOR PERSONAL EGOCENTRIC VIDEO RETRIEVAL

Conventional egocentric retrieval (*e.g.,* TFR CVR) uses a two-stage pipeline to form the target description, which can lose visual detail and dilute user-specific cues. We introduce `EgoRetriever`, a one-stage, training-free framework for long context personal retrieval. It leverages an MLLM to directly produce a detailed target description conditioned on the query, personal memory, and a lightweight visual anchor, without additional training. Formally, with MLLM $\Psi_M$,

$$T_t = \Psi_M(p_c \circ Q \circ \mathcal{M}_q \circ I_r). \tag{1}$$

The reflective CoT prompt $p_c$ (Figure 2) guides single prompt reasoning with three concise stages (full template in Appendix A.2.2):

**Reference Image Description.** During this initial step, the MLLM provides a detailed description of the visual content relevant to the user's query. In Figure 2, irrelevant elements such as general room features (*e.g.,* coffee table) are selectively omitted, while intention-relevant elements (*e.g.,* a black and white border collie standing beside a brown leather sofa, user's hand petting the dog) are preserved to clearly align with the user's retrieval intent.

**Thoughts.** Given the relevant visual details and the user's query, the MLLM interprets the implicit retrieval intent. The MLLM explicitly reasons about which visual and contextual cues most significantly influence its understanding of the user's query. Specifically, it identifies critical visual attributes (*e.g.,* the black and white border collie) and contextual information from the personalized memory bank (*e.g.,* frequent interaction involving a man and a sofa). This reasoning aligns the visual cues with memory patterns, guiding the inference toward the user's probable interaction partner.

**Reflections.** Given the potential intentions and reference image, the MLLM filters the inferred intentions by explicitly considering the coherence and context of visual and textual details. Potential irrelevant assumptions (*e.g.,* interactions unrelated to the query context) are excluded. The MLLM clarifies the rationale behind identifying a man interacting with the user and the border collie near a brown sofa, thereby reducing ambiguity and hallucinations.

**Target Video Description.** Finally, given the filtered intentions and relevant visual-contextual reasoning results, the MLLM generates the accurate target video description. This description succinctly captures the user's intended interaction (*e.g.,* the user and a man interacting with the black and white border collie on a brown leather sofa), clearly matching the user's original retrieval query.

After generating the target description $T_t$, `EgoRetriever` uses a video-language retrieval backbone (*e.g.*, EgoVLPv2 Pramanick et al. (2023)) to identify the most relevant video clips from a candidate pool. $T_t$ is encoded using a pretrained text encoder $\Psi_T$, and each candidate clip $C_i$ is embedded using a video encoder $\Psi_V$. The most relevant clip $C^*$ is then obtained via cosine similarity:

$$C^* = \underset{C_i \in \mathcal{C}}{\operatorname{argmax}} \frac{\Psi_V(C_i)^\intercal \Psi_T(T_t)}{\|\Psi_V(C_i)\|\|\Psi_T(T_t)\|}. \tag{2}$$

## 4 EXPERIMENTS

### 4.1 EGOMEMORY BENCHMARK

To rigorously evaluate *personalized* egocentric video retrieval, we introduce the **EgoMemory** benchmark, derived from Ego4D's Natural Language Queries (NLQ) Grauman et al. (2022). NLQ provides ∼227 hours of head-mounted video from 137 participants across 74 locations, annotated with free-form queries about "when/where/with whom/what," reflecting realistic recall scenarios.

**Limitations of Ego4D's NLQ for Personalization.** While rich and diverse, NLQ was designed for temporal localization within *individual* clips and does not aggregate multi-video, user-specific context. In particular, videos are not grouped by user at retrieval time, ownership or user-linkage cues are not modeled, and queries need not require long-horizon evidence. Consequently, evaluating memory-augmented personalized retrieval directly on NLQ risks rewarding scene/frequency priors rather than genuine personalization.

**Design of EgoMemory: user grouping and query filtering.** To address this, we treat each user as a distinct retrieval unit by aggregating *all* of their videos as personal context at query time. From 137 participants, we manually select 45 with sufficient temporal coverage to emulate practical AR usage. To ensure that evaluation genuinely probes personalization *and* long-context reasoning, we apply a two-stage filtering pipeline: (i) **GPT-4o pre-screening** to keep queries with explicit personal references or strong user linkage (*e.g.,* possessives, deictic cues) and to discard queries designed for short single-clip answers; (ii) **Human verification** to confirm that the main referenced object is plausibly user-linked across history. Concretely, for each retained query–target pair, annotators review 20 additional short clips from the same user containing the same object class; queries are labeled *personal* when $\geq 90\%$ of reviewed instances match the target and *uncertain* when $\geq 75\%$. This results in $\sim$91.6% personal queries. The resulting dataset comprises 245 videos from 45 users and 639 queries. For details, please refer to the Appendix A.3.

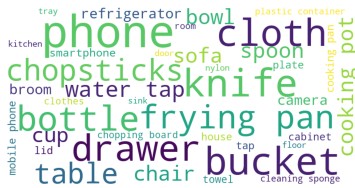

Figure 3: The most frequently objects in constructed memory banks.

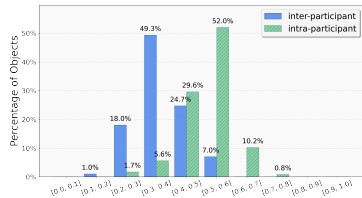

Figure 4: Similarity of top 100 objects in constructed memory banks.

**Memory banks and added annotations.** For each participant, we annotate 165,795 user-specific object annotations across 12 attributes (category, color, texture, shape, brand, state, etc.) for personal memory bank construction (Section 3.2). The number of unique object types per user ranges from 59 to 638 (median 129); memory sizes range from 322 to 10,454 entries (median 1,312). This structured representation goes beyond unstructured transcripts by explicitly modeling user-specific contextual cues (*i.e.,* recurrence + linguistic cues) and supports training-free, interpretable personalization.

**Personalization Heterogeneity.** To quantify user specificity, we compute Jaccard similarity over attribute sets for the 100 most frequent object types, comparing inter- vs. intra-participant distributions. Also, to account for environment bias, we recompute inter-participant similarity, restricting comparisons to matched coarse scenes (kitchen, living room, outdoor) using the "Status" metadata, i.e., $J(A^{\text{scene}=s}, B^{\text{scene}=s}) = \frac{|A \cap B|}{|A \cup B|}$. As shown in Figure 4, **68.3%** of objects $< 0.4$, confirming that personalization persists beyond scene priors.

**Candidate set for retrieval.** The candidate retrieval pool contains 2,228 clips from participants' histories (mean $\sim$33 clips/user), spanning 4 to 300 s (mean 103.82 s), totaling 64.25 h. By construction, selected queries target objects recurring across a user's videos, so relevant evidence often lies outside the target clip. This constitutes a *long-context* setting that requires integrating $(Q, \mathcal{M}_q, I_r)$ with cross-video evidence rather than relying on single-clip shortcuts.

**Evaluation Metrics.** We adopt **mean Recall@K** across users as our principal evaluation metric, reporting mean Recall@1, mean Recall@2, and mean Recall@3. Specifically, for each user, we compute Recall@K based on their individual candidate set and then average across all users to obtain a macro-level performance summary. This approach ensures fair contribution from each user, mitigating the bias that could arise from varying query counts per user. Similar evaluation metrics are also adopted in Ego4D Episodic Memory benchmarks Grauman et al. (2022) and EgoCVR Hummel et al. (2024). Moreover, in settings with a single answer per query, Recall@K is equivalent to Hit Rate@K, widely accepted in recommender systems Sun et al. (2019). Candidate set statistics are provided in the Appendix A.9.

**Implementation Details.** We leverage GPT-4o for constructing the user-specific memory banks by generating detailed object-centric metadata from video clips. Additionally, GPT-4o is also employed to perform reflective CoT reasoning. Our retrieval experiments were conducted using four NVIDIA V100 GPUs with 32GB each. We evaluated multiple state-of-the-art video-language models, including LanguageBind Zhu et al. (2023), CLIP Radford et al. (2021), BLIP Li et al. (2022), and EgoVLPv2 Pramanick et al. (2023). We employ EgoVLPv2 as the text encoder in `EgoRetriever`. CLIP and BLIP visual representations for the videos are obtained by averaging embeddings from 15 uniformly sampled image frames. For each candidate video, visual embeddings were extracted and subsequently matched with the GPT-4o-generated textual descriptions via cosine similarity. Please refer to the Appendix A.10 for more details.

Table 1: Mean Recall@K (%) for different retrieval configurations. "Video Model" indicates use of a video encoder, "Textual Memory Bank" denotes personal text-based memory, "Visual Reference" represents visual info in the reference image, and "Fusion Strategy" specifies modality combination ("Avg" refers to naive average fusion). The best and second-best results are in **bold** and underlined.

| Method | Video Model | Textual Memory Bank | Visual Reference | Fusion Strategy | Mean Recall (%) mR@1 | mR@2 | mR@3 |
|---|---|---|---|---|---|---|---|
| Random | ✗ | ✗ | ✗ | — | 3.62 | 9.74 | 15.23 |
| CLIP | ✗ | ✓ | ✗ | — | 10.41 | 12.95 | 16.72 |
| BLIP | ✗ | ✓ | ✗ | — | 10.88 | 13.67 | 17.48 |
| EgoVLPv2 | ✓ | ✓ | ✗ | — | 11.25 | 15.03 | 18.30 |
| LanguageBind | ✓ | ✓ | ✗ | — | 11.02 | 14.60 | 17.83 |
| CLIP | ✗ | ✗ | ✓ | Avg | 14.74 | 17.53 | 21.41 |
| BLIP | ✗ | ✗ | ✓ | Avg | 15.12 | 18.07 | 22.82 |
| EgoVLPv2 | ✓ | ✗ | ✓ | Avg | 15.77 | 20.61 | 23.79 |
| LanguageBind | ✓ | ✗ | ✓ | Avg | 15.26 | 20.14 | 23.04 |
| CLIP | ✗ | ✓ | ✓ | Avg | 15.64 | 18.63 | 22.71 |
| BLIP | ✗ | ✓ | ✓ | Avg | 16.02 | 19.17 | 24.12 |
| EgoVLPv2 | ✓ | ✓ | ✓ | Avg | 16.67 | 21.71 | 25.09 |
| LanguageBind | ✓ | ✓ | ✓ | Avg | 16.16 | 21.24 | 24.34 |
| BLIP$_{CoVR}$ Ventura et al. (2024) | ✗ | ✓ | ✓ | Cross-Attn. | 15.94 | 19.17 | 23.00 |
| BLIP$_{CoVR-ECDE}$ Thawakar et al. (2024) | ✗ | ✓ | ✓ | Cross-Attn. | 16.41 | 19.63 | 23.64 |
| CIReVL Karthik et al. (2024) | ✗ | ✓ | ✓ | Captioning | 16.95 | 20.13 | 24.37 |
| OSrCIR Tang et al. (2024a) | ✗ | ✓ | ✓ | Captioning | 17.28 | 21.64 | 25.49 |
| TFR-CVR Hummel et al. (2024) | ✓ | ✓ | ✓ | Captioning | 18.21 | 27.12 | 32.05 |
| EgoRetriever (Ours) | ✓ | ✓ | ✓ | Captioning | **23.19** | **38.48** | **47.83** |

## 4.2 MAIN RESULTS

We compare `EgoRetriever` to three families of egocentric retrieval systems: (i) *Training free encoders*, **CLIP** Radford et al. (2021), **BLIP** Li et al. (2022), **EgoVLPv2** Pramanick et al. (2023), and **LanguageBind** Zhu et al. (2023), with video features from 15 uniformly sampled frames, evaluated under three input regimes: (a) query plus textual memory, (b) query plus visual reference, and (c) late fusion of all three. (ii) *Composed image retrieval*, **BLIP$_{CoVR}$** Ventura et al. (2024), **BLIP$_{CoVR-ECDE}$** Thawakar et al. (2024), **CIReVL** Karthik et al. (2024), and **OSrCIR** Tang et al. (2024a), which generate a target description with an LLM and retrieve, typically via CLIP. (iii) *Egocentric aware* **TFR-CVR** Hummel et al. (2024), which captions a key frame and prompts an LLM to form the target description before first-person video retrieval. Unlike these two-stage pipelines, `EgoRetriever` performs one-stage reflective reasoning within an MLLM to directly produce the target description. Both TFR-CVR and `EgoRetriever` use the same MLLM (GPT 4o), and all baselines receive identical inputs and encoders, ensuring fair comparison.

Table 1 reports mean Recall@K performance for various retrieval configurations on the EgoMemory benchmark. Our `EgoRetriever` achieves the best performance across all metrics, with a notable mR@1 of 23.19% and mR@3 of 47.83%, outperforming the best egocentric-aware baseline (TFR-CVR) by 4.98% and 15.78%, respectively. This significant improvement underscores the effectiveness of our reflective CoT prompting and personalized memory bank design for long-context, user-centric retrieval. Ablation across modality configurations demonstrates that models relying solely on textual memory are limited in capturing user intent (*e.g.,* EgoVLPv2 mR@1: 11.25%), highlighting the necessity of integrating visual references. The combination of textual memory and visual reference leads to consistent gains, with our approach delivering the highest recall even in complex, diverse scenarios. Compared to state-of-the-art composed image retrieval models, `EgoRetriever` yields a substantial relative gain, improving mR@1 by 5.91% over OSrCIR and nearly doubling mR@3 performance. These results validate the value of leveraging rich, user-specific historical data and long-context modeling: by constructing comprehensive personal memory banks from aggregated user histories (*i.e.,* with thousands of annotated attributes per user), our framework enables long-context reasoning over user context, habits, and object interactions that are critical for accurate retrieval. For more qualitative analysis, please refer to the Appendix A.2.4.

## 4.3 ABLATION STUDY AND PERFORMANCE ANALYSIS

In Table 2, we assess the contribution of each component on EgoMemory. **(1) Models '2–7' evaluate the necessity of key modules within `EgoRetriever`.** Removing textual memory yields the largest drop in mean Recall (model '2') by 7.94% compared to our full model ('1'), underscoring

Table 2: Results in terms of mR@1, mR@2, and mR@3 demonstrating the necessarily of the key modules in `EgoRetriever`.

|  | Methods | mR@1 | mR@2 | mR@3 |
|---|---|---|---|---|
| 1. | Full Model (GPT-4o) | 23.19 | 38.48 | 47.83 |
| **Significance of key modules of our `EgoRetriever`** | | | | |
| 2. | w/o Textual Memory | 18.04 | 30.73 | 36.91 |
| 3. | w/o Reference Frame | 20.79 | 32.94 | 40.23 |
| 4. | w/o Original Description | 21.49 | 36.04 | 43.70 |
| 5. | w/o Thoughts | 20.14 | 33.62 | 41.89 |
| 6. | w/o Reflection | 20.52 | 35.17 | 42.90 |
| 7. | w/o ICT | 21.37 | 36.29 | 43.49 |
| **Impact of different CoT methods** | | | | |
| 8. | Simple CoT | 20.13 | 33.18 | 41.28 |
| 9. | Advance CoT | 19.42 | 32.50 | 41.02 |
| **Practical feasibility without human-written narrations** | | | | |
| 10. | Human captions | 23.19 | 38.48 | 47.83 |
| 11. | EgoGPT auto-captions | 22.73 | 36.72 | 46.02 |
| 12. | GPT-4o auto-captions | 21.19 | 35.22 | 45.39 |
| **Impact of different MLLMs** | | | | |
| 13. | LLaVA | 20.37 | 33.58 | 41.90 |
| 14. | Qwen2.5-VL | 22.03 | 35.24 | 45.27 |
| 15. | GPT-4o-mini | 22.31 | 37.19 | 46.43 |

Table 3: Results that emphasize the importance of our personal memory bank.

| Method | Memory Structure | mR@1 | mR@2 | mR@3 |
|---|---|---|---|---|
| TFR-CVR | GPT-4o Caption | 11.41 | 16.57 | 20.04 |
| | EgoGPT Caption | 13.09 | 19.37 | 21.32 |
| | VideoAgent | 13.09 | 19.37 | 21.32 |
| | Metadata | 18.21 | 27.12 | 32.05 |
| EgoRetriever | GPT-4o Caption | 13.18 | 19.30 | 21.17 |
| | EgoGPT Caption | 15.31 | 20.57 | 26.74 |
| | VideoAgent | 17.49 | 26.40 | 35.62 |
| | Metadata | 23.19 | 38.48 | 47.83 |

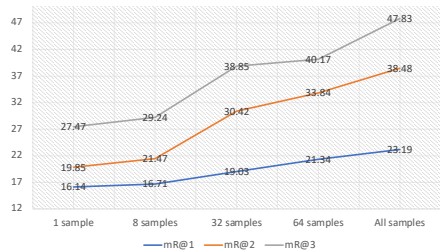

Figure 5: Effect of the number of metadata for each object in the memory bank.

the need for user-linked metadata. Similarly, the absence of the reference frame (model '3') leads to a 5.18% drop, emphasizing its critical role in grounding the visual context. Within the reflective CoT, omitting original description, thoughts, or reflection reduces performance by roughly 3%–5%, and removing ICT examples gives a smaller but consistent decline. Together, these results show that memory and a lightweight visual anchor carry most of the gain, while each CoT step contributes additive improvements. **(2) Models '8–9' compare our Reflective CoT against other CoT methods.** Replacing our reflective CoT with simple CoT or an advanced two-stage CoT (DDCoT) degrades mean Recall by about 5%, indicating the advantage of single prompt reflective reasoning for interpreting multimodal user intent. **(3) Models '10–12' examine the practical feasibility without human-written narrations.** We re-annotated all 165,795 objects using EgoGPT and GPT-4o captions for each reference frames. Compared to human captions (model '10'), EgoGPT auto captions (model '11') and GPT-4o auto captions (model '12') show only minor declines, confirming that `EgoRetriever` remains effective without ground truth narrations and is practical for real-world deployment. **(4) Models '13–15' examine the impact of different MLLMs on performance.** Utilizing open-source MLLMs such as LLaVA Liu et al. (2023) (model 13') and Qwen2.5-VL Yang et al. (2024) (model 14') achieves competitive but clearly inferior results compared to GPT-4o, with performance gaps of 4.55% and 2.32%, respectively. Notably, GPT4o-mini (model '15') performs closely to GPT-4o, with only a minor decline of 1.19%, indicating that GPT4o-mini offers a promising balance between efficiency and retrieval performance.

## 4.4 ANALYSIS

In this subsection, we provide detailed analyses of our design choices and the common failure cases.

**Analysis of Memory Bank Design.** Table 3 reports retrieval performance under different memory bank structures. Across both TFR-CVR and `EgoRetriever`, our structured metadata yields clear gains over caption memories from GPT–4o and EgoGPT. For TFR-CVR, metadata improves average performance by 9.79% and 7.87% relative to GPT-4o and EgoGPT, respectively. For `EgoRetriever`, the gains are larger at 18.62% and 15.63%. We also evaluate the VideoAgent memory bank Fan et al. (2024), which uses a temporal caption memory and an object memory, and find it trails our structured metadata and is comparable to EgoGPT for TFR-CVR. We attribute the advantage to storing 12 attribute fields and frequency statistics that emphasize recurrent personal objects, enabling richer long-context grounding than caption-only or non-personalized memories.

**Generalization ability of our `EgoRetriever`.** We evaluate `EgoRetriever` on EgoCVR Hummel et al. (2024), which similarly requires generating a target video description from a reference image and a textual edit; following protocol, we report both *global* and *local* search. As shown in Table 4, `EgoRetriever` achieves the best results in all settings, surpassing the two stage TFR-CVR*

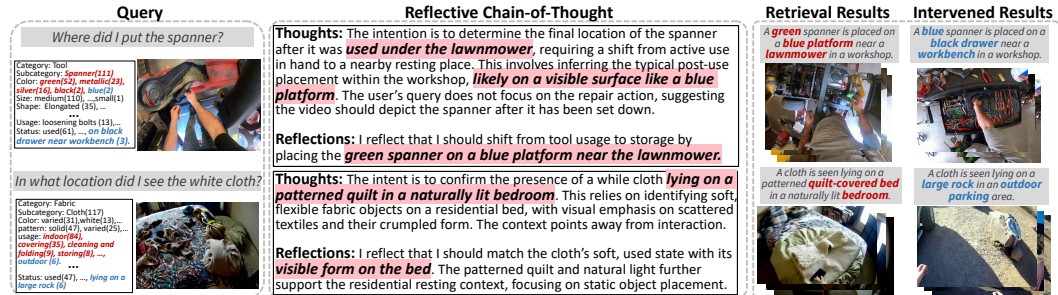

Figure 6: Visualization of common failure cases. The red color denotes the incorrect reasoning outcomes of intention. The top-1 retrieval result and the intervened correction are shown.

baseline (GPT-4o as captioner and LLM) by an average of 5.05%. We further test transfer to Ego-LifeQA Yang et al. (2025), a fixed answer QA suite probing entity logs, past event recall, and habit patterns, where `EgoRetriever` obtains SoTA, with average performance gains of 6.17% over the EgoGPT. Together, these results demonstrate strong generalization of our one-stage, training-free framework beyond EgoMemory. For details, please refer to the Appendix A.4,A.5.

**Impact of Memory Bank Context Length.** We evaluate the influence of memory bank context length by varying the number of included object metadata, as shown in Figure 5. Our results reveal that restricting the memory bank to short-term contexts significantly limits retrieval performance, as essential long-term user patterns and object interactions are underrepresented. As the context length increases, incorporating a broader history of user experiences, retrieval accuracy improves markedly. Notably, substantial performance gains are observed when extending the memory bank to encompass more metadata, highlighting the importance of long-context information for modeling fuser semantics. These findings underscore that a comprehensive, extensive memory bank is crucial for enabling accurate and personalized long-context video retrieval.

Table 4: Generalization of EgoRetriever on EgoCVR and EgoLifeQA.

**EgoCVR** Hummel et al. (2024)

| Method | Global | | | Local | | |
|---|---|---|---|---|---|---|
| | R@1 | R@5 | R@10 | R@1 | R@2 | R@3 |
| CIReVL | 2.0 | 6.8 | 10.6 | 33.6 | 49.7 | 61.4 |
| OSrCIR | 4.9 | 9.3 | 13.4 | 37.4 | 53.3 | 68.1 |
| TFR-CVR | 14.1 | 39.5 | 54.4 | 44.2 | 61.0 | 73.2 |
| TFR-CVR* | 14.7 | 41.2 | 55.6 | 46.1 | 62.4 | 73.9 |
| EgoRetriever | **17.4** | **49.2** | **62.7** | **50.3** | **68.2** | **76.4** |

**EgoLifeQA** Yang et al. (2025)

| Method | EntityLog | EventRecall | HabitInsight |
|---|---|---|---|
| GPT-4o | 34.4 | 42.1 | 29.5 |
| LLaVA-OV | 36.8 | 34.9 | 31.1 |
| EgoGPT | 39.2 | 36.5 | 31.1 |
| EgoRetriever | **42.5** | **45.1** | **37.7** |

**Analysis of Common Failure Cases.** To assess the limitations of `EgoRetriever`, we examined 100 failure cases. As shown in Figure 6, we identify two primary issues: (1) *Object Disambiguation Challenges* (74%): The model often fails to accurately distinguish target objects in cluttered scenes, such as identifying a specific spanner among visually similar tools in a workshop (Row 1). (2) *Context Misinterpretation* (21%): The model may misinterpret user intent when the visual reference context is ambiguous. For example, retrieving an indoor scene for a "white cloth" when the correct context is outdoors (Row 2). Notably, supplementing queries with more detailed contextual cues (*e.g., black drawer, outdoor*) can mitigate these errors, highlighting the need for enhanced object differentiation and context reasoning in personalized, long-context egocentric video retrieval.

## 5 CONCLUSION

In this paper, we tackle the challenge of personalized, long-context episodic memory retrieval from egocentric video. We introduce EgoMemory, a benchmark built from Ego4D with user-specific memory banks and diverse, context-rich queries. We further propose `EgoRetriever`, a training-free framework leveraging Multimodal Large Language Models and reflective Chain-of-Thought reasoning to explicitly understand user queries through personal memory for personalized video retrieval. Extensive experiments on EgoMemory and EgoCVR demonstrate that our approach achieves state-of-the-art performance and strong generalization, marking a significant advance in practical personalized and long-context egocentric video retrieval. It inspires future research on user-centric memory augmentation and has broad implications for real-world multimodal AI applications.

## ETHICS STATEMENT

**Scope and alignment with the ICLR Code of Ethics.** Our work follows the ICLR principles of responsible stewardship: contributing to well-being, upholding scientific excellence, avoiding harm, being honest and transparent, ensuring fairness and non-discrimination, respecting prior work, respecting privacy, and honouring confidentiality.

**Human data, consent, and provenance.** All experiments use public egocentric datasets (Ego4D, EgoCVR, EgoLifeQA) that were collected under their own consent and governance processes. We introduce no new human data collection and do not attempt re-identification or linkage to external records. We comply with dataset licenses and intended use policies and acknowledge all sources.

**Privacy by design.** Our method is training-free and centers on personalization of the device. The personal memory bank is generated and stored locally during idle periods, giving users control over creation, inspection, and deletion. At query time, only minimal structured metadata (attribute tuples and frequency counts) is shared with the language model. Raw video, audio, faces, names, locations, and other directly identifying content are not transmitted. Open source MLLMs can replace hosted services for fully local inference when stricter privacy is required.

**Transparency, reproducibility, and integrity.** We report methods, prompts, model choices, and evaluation protocols to support replication. We do not fabricate or obfuscate results, and we disclose limitations and failure modes. If released, code and de-identified annotations will include documentation of data provenance and usage constraints.

**Fairness and potential harms.** Egocentric data can encode social and environmental biases. Our memory construction focuses on object-level attributes rather than demographic attributes, and we report per benchmark generalization. We discourage use in surveillance or monitoring of individuals. Any released resources will carry a license that prohibits re-identification, profiling, law enforcement, or use targeting protected classes.

**Confidentiality and data handling.** We do not handle confidential third-party data. When hosted MLLMs are used, we enable no retention settings where available and minimize payloads. Access credentials are stored outside the artifacts released for research.

**Annotator well-being.** Manual verification was performed by trained team members following internal guidelines. Annotators could skip any item, and no harmful content was intentionally introduced.

**Environmental impact.** We emphasize efficiency through precomputed embeddings and single-stage reasoning. Measured query latency and memory remain stable as data volume grows, which reduces energy cost and enables deployment on modest hardware.

**Limitations and remediation.** A formal privacy guarantee, such as differential privacy, is out of scope and is an important direction for future work. We welcome community feedback and will follow ICLR processes for raising concerns and remediation if any ethical issues are identified.

## REPRODUCIBILITY STATEMENT

We took several steps to ensure that our results are reproducible. An anonymized repository included in the supplementary materials provides training/inference scripts, configuration files, environment specifications, and pretrained checkpoints for the backbones used in our experiments, along with a minimal working example and sample data to verify end-to-end execution. The full method is specified in the main text (Section.3,4), with the complete algorithmic procedure and prompt templates given in Appendix A.2.1–A.2.3 and qualitative analyses in A.2.4; the personalized memory bank design, construction template, and cross-user similarity algorithm are detailed in A.1.1–A.1.2 with additional qualitative analysis in A.1.3. Dataset usage and evaluation details are organized per benchmark: EgoMemory (A.3), EgoCVR (A.4), and EgoLifeQA (A.5), including task definitions, preprocessing, official protocols, and metric definitions. We report ablations and sensitivity studies in A.6, and provide compute/runtime profiling and latency breakdowns in A.7–A.8. Candidate set statistics are summarized in A.9, and further implementation notes (e.g., seed control, batching, precision settings) appear in A.10. To sum, these materials are intended to allow independent researchers to reproduce the key results reported in the paper with minimal additional assumptions.

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

# A APPENDIX

## TABLE OF CONTENTS FOR APPENDIX

## A.1 DETAILS FOR OUR PERSONALIZED MEMORY BANK

### A.1.1 COMPLETE TEMPLATE FOR PERSONALIZED MEMORY BANK CONSTRUCTION

In constructing the personalized memory bank, we leverage a structured prompt that guides a pretrained multimodal large language model (MLLM) to systematically extract detailed object attributes from textual narrations and visual contexts, as outlined in Figure 7. The prompt explicitly instructs the model to identify and describe the primary object interacted with by the user, ensuring structured output consistency in JSON format. This structured extraction facilitates precise aggregation and retrieval of personalized attributes critical for the memory bank.

**Structured JSON Format.** The prompt mandates the generation of a strictly structured JSON object with explicitly defined top-level keys (*e.g.,* major category, subcategory, color, texture, shape, material, brand, style, pattern, feature, usage, status). Each key corresponds to a specific attribute dimension necessary for capturing fine-grained personal contextual details, ensuring uniformity and ease of downstream processing.

You are a helpful vision assistant that identifies object attributes in a structured JSON format. Ensure your output is valid JSON with exactly the specified top-level keys.

Please carefully analyze the following sentence and identify the primary object being interacted with by the speaker.
The sentence is: "{object_sentence}"

Consider both the sentence and, if applicable, any associated visual information to determine the object and its attributes.

Your task is to describe this object's attributes in a structured JSON format.
The JSON output must contain EXACTLY the following top-level keys:
{required_keys_str} # Dynamically populated list, e.g., major_category, subcategory, color, texture, shape, material, brand, style, pattern, feature, usage, status.

## Guidance for attribute values
1. Be Specific and Descriptive: For each attribute, provide the most accurate and detailed value you can infer. For example, for 'color', if an object is "dark blue with yellow stripes", please state that rather than just "blue" or "patterned".
2. Use Known Values as Examples: Below is a list of attribute categories and examples of values seen previously. Use these to understand the type of information expected. If a relevant value is present, you can use it.
    Known attribute examples:
{known_str_joined} # Dynamically populated, e.g., "- color: (e.g., red, blue, green, ...)"
3. Invent New Values When Necessary: If the object has a characteristic not covered by the examples or if the examples are not relevant, provide a new, concise, and descriptive value. This is how we discover new attributes.
4. Handling Uncertainty/Not Applicable: If an attribute's value cannot be determined from the provided information or is not applicable to the object, use "unspecified" or "N/A". Avoid guessing if confidence is very low. For 'brand', if not explicitly mentioned or visible, "unspecified" is appropriate.
5. Compound Attributes: For attributes like 'material' (e.g., "fabric and wood") or 'feature' (e.g., "supporting and sleeping"), list the distinct components as a single string.
6. Consistency: Ensure your entire output is a single, valid JSON object with all string values properly escaped.

Example of desired thinking for 'color' if a bed is blue and white patterned:
"color": "blue and white patterned"

Now, based on the sentence "{object_sentence}" and any visual context, provide the JSON output.

Figure 7: The complete template of our personal memory construction.

**Specificity and Descriptive Precision.** To maximize the accuracy and richness of the memory bank, the prompt instructs the model to provide detailed, descriptive attribute values. It explicitly discourages vague descriptions, advocating specificity, for instance, specifying "dark blue with yellow stripes" rather than a generic label like "blue" or "patterned." This precision enhances the utility of the memory bank for fine-grained retrieval.

**Known Values and Inventive Flexibility.** The prompt includes dynamically populated examples of known attribute values, providing clarity and consistency in expected responses. However, recognizing the inherent novelty in egocentric video contexts, the prompt encourages the model to introduce new, concise, and descriptive attribute values when existing examples are insufficient, thereby continually enriching the attribute taxonomy.

---

**Algorithm 1** Calculating Average Diversity for an Object Class

---

**Input**: An object class $O$, a global attribute database $\mathcal{D}$ mapping each user $U_k$ to their attribute set $A_k$ for object $O$.

**Output**: Average diversity score $\text{Div}_{\text{avg}}(O)$ for object class $O$.

1: Let $\mathcal{U}_O = \{U_k \mid (U_k, A_k) \in \mathcal{D} \text{ for object } O\}$ be the set of all users with attributes for object $O$.

2: Initialize a list of similarity scores $S = []$.

3: **for** each unique pair of users $(U_i, U_j)$ in $\mathcal{U}_O$ where $i \neq j$ **do**

4:     Retrieve attribute sets for each user: $A_i = \mathcal{D}(U_i, O)$ and $A_j = \mathcal{D}(U_j, O)$.

5:     Compute the Jaccard similarity:

$$s_{ij} = J(A_i, A_j) = \frac{|A_i \cap A_j|}{|A_i \cup A_j|}$$

6:     Add $s_{ij}$ to $S$.

7: **end for**

8: **if** $S$ is not empty **then**

9:     Compute the average similarity: $\text{Sim}_{\text{avg}}(O) = \frac{1}{|S|} \sum_{s \in S} s$

10:     Compute the average diversity: $\text{Div}_{\text{avg}}(O) = 1 - \text{Sim}_{\text{avg}}(O)$

11: **else**

12:     Set average diversity to 0 (or undefined if only one user has the object): $\text{Div}_{\text{avg}}(O) = 0$

13: **end if**

14: **return** $\text{Div}_{\text{avg}}(O)$

---

**Handling Uncertainty.** To maintain reliability and mitigate incorrect assumptions, the prompt explicitly instructs the model to use "unspecified" or "N/A" when an attribute value cannot be confidently determined from available information. This approach preserves the integrity and trustworthiness of the memory bank by avoiding low-confidence guesses.

**Compound Attributes and Consistency.** The prompt clearly addresses compound attributes (e.g., combining "fabric and wood" for material attributes), requiring these to be succinctly represented as unified strings. Additionally, it underscores consistency across responses, ensuring that all outputs adhere strictly to valid JSON formatting with appropriate string escaping. This structured consistency facilitates seamless integration into the personalized memory bank infrastructure.

Collectively, these explicit instructions ensure the prompt's effectiveness in systematically extracting detailed, personalized attributes crucial for constructing a robust and reliable personalized memory bank for egocentric video retrieval.

### A.1.2 ALGORITHM OF CALCULATING CROSS-USER OBJECT ATTRIBUTE SIMILARITY

To quantitatively evaluate the attribute diversity of object classes across different users within our proposed personal memory bank, we introduce a systematic method based on attribute similarity metrics. Specifically, we employ the **Jaccard index**, a widely recognized measure for quantifying similarity between finite attribute sets. The discrete and non-hierarchical nature of our object attribute metadata makes the Jaccard index particularly suitable for this analysis.

Algorithm 1 formally describes the calculation procedure. For each object class, the algorithm computes pairwise Jaccard similarity scores between attribute sets associated with every unique pair of users who interact with the same object. Subsequently, the algorithm derives an aggregate diversity score by taking the complement of the average of these pairwise similarities. This aggregate metric, termed the *average diversity*, intuitively captures the heterogeneity in how users characterize identical objects in their personalized memory banks. A higher average diversity score explicitly indicates a richer variability in user-specific object descriptions, underscoring the contribution of our personalized memory bank design toward enhanced contextual representation.

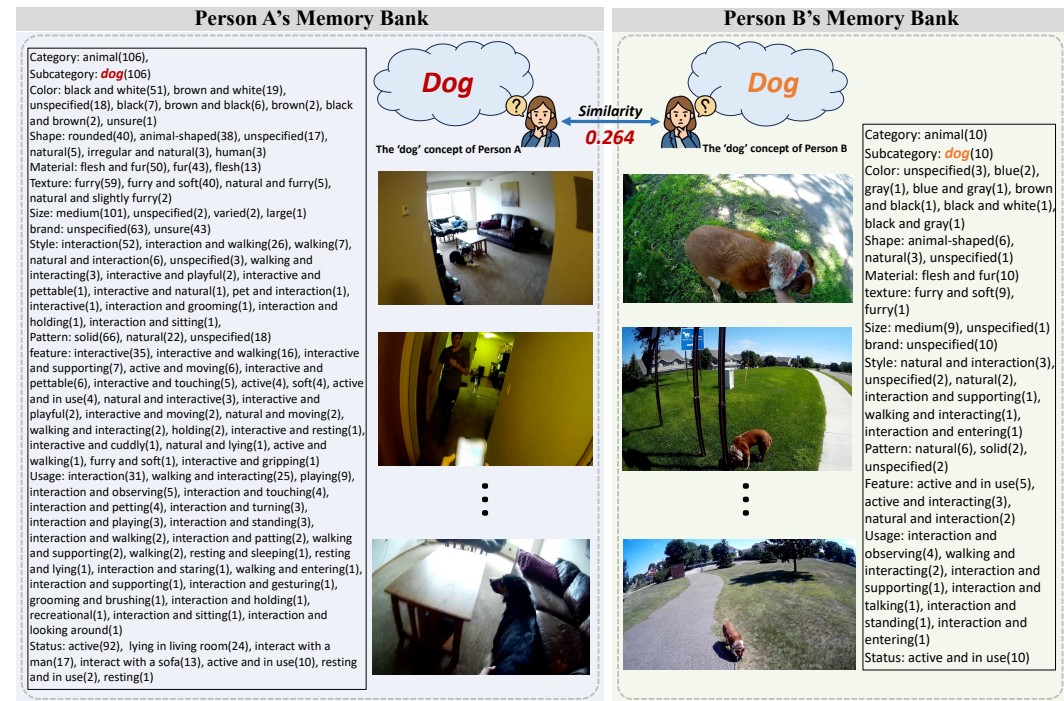

Figure 8: Qualitative comparison of the object concept "dog" across two users' personal memory banks. Attribute distributions reveal substantial divergence in visual, contextual, and interactional properties. The computed similarity score (Jaccard index) is **0.264**, highlighting the necessity of modeling user-specific memory to capture personalized object semantics. .

### A.1.3 Qualitative Analysis for Personalized Memory Bank

To better understand the practical implications of our proposed personalized memory bank, we conducted a qualitative analysis comparing attribute representations of identical object categories across different users. As an illustrative example in Figure 8, we examined the concept of "dog" as represented by two distinct users (Person A and Person B), each with their own historical interactions captured within their respective personal memory banks.

The detailed attribute annotations reveal substantial variations between the two users in aspects such as color, pattern, style, usage, and specific interaction contexts. Quantitatively, the computed similarity score between these two users' personal memory for the concept of "dog" (Indoor *vs* Outdoor) is notably low, at **0.264**. This underscores a significant divergence in their individual conceptualizations and experiences associated with the same general object class.

The low similarity score highlights a crucial insight: object attributes are perceived and recalled uniquely by different users based on their personal experiences and contexts. Thus, it clearly demonstrates the necessity and importance of constructing personalized memory banks, as generic or aggregated memory representations would inadequately capture the rich variability in individual user interactions and perceptions. Our findings reinforce the core contribution of our personalized memory bank framework, its ability to accurately reflect nuanced user-specific memory contexts, ultimately enhancing personalized retrieval performance.

## A.2 Details for Our EgoRetriever's Process

### A.2.1 Algorithm of EgoRetriever's Process

Algorithm 2 outlines the comprehensive procedure of EgoRetriever for training-free, long-context personal egocentric video retrieval. The process initiates with a natural language query $Q$, a candidate set of video clips $\mathcal{C}$, and a user-specific semantic memory bank $\mathcal{M}$. EgoRetriever first

---

**Algorithm 2** Computing Process of `EgoRetriever` for Personal Egocentric Video Retrieval

---

**Input**: Natural language query $Q$, candidate clip set $\mathcal{C} = \{C_1, C_2, \ldots, C_M\}$, user-specific semantic memory bank $\mathcal{M}$, reflective CoT prompt $p_c$.
**Parameters**: Frozen Multimodal Large Language Model (MLLM) $\Psi_M$, frozen text encoder $\Psi_T$, frozen video encoder $\Psi_V$.
**Output**: Retrieved target clip $C^*$.

 1: Initialize pre-trained and frozen models $\Psi_M, \Psi_T, \Psi_V$.
 2: Retrieve personal context from memory bank: Query $\mathcal{M}$ with $Q$ to obtain personal object metadata $\mathcal{M}_q$ and reference image $I_r$.
 3: Generate target clip description using reflective CoT:

$$T_t = \Psi_M(p_c \circ Q \circ \mathcal{M}_q \circ I_r)$$

 4: Compute normalized text embedding for the target description:

$$\hat{e}_T = \frac{\Psi_T(T_t)}{\|\Psi_T(T_t)\|_2}$$

 5: Initialize a list of similarity scores $S = []$.
 6: **for** each candidate clip $C_i$ in $\mathcal{C}$ **do**
 7:     Compute normalized video embedding for the candidate clip:

$$\hat{e}_{V_i} = \frac{\Psi_V(C_i)}{\|\Psi_V(C_i)\|_2}$$

 8:     Compute similarity score: $s_i = \hat{e}_{V_i}^\top \hat{e}_T$ {Cosine similarity for normalized embeddings}
 9:     Add $s_i$ to $S$.
10: **end for**
11: Retrieve target clip: $C^* = \underset{C_i \in \mathcal{C}}{\operatorname{argmax}} \, S_i$ {Select clip with highest similarity score}
12: **return** $C^*$

---

leverages the MLLM $\Psi_M$ to consult the memory bank $\mathcal{M}$ and retrieve pertinent personal object metadata $\mathcal{M}_q$ and a reference frame $I_r$. Subsequently, using these contextual cues along with the original query $Q$ and a reflective CoT prompt $p_c$, a detailed target clip description $T_t$ is generated. This description $T_t$ is then encoded using a text encoder $\Psi_T$. Each candidate clip $C_i \in \mathcal{C}$ is encoded using a video encoder $\Psi_V$. The final retrieval of the target clip $C^*$ is achieved by computing the cosine similarity between the encoded description and each encoded candidate clip. This approach allows for a modular retrieval pipeline where the core reasoning and description generation are handled by the MLLM, independent of the specific video-language encoders used, requiring no additional training.

### A.2.2   COMPLETE TEMPLATE FOR REFLECTIVE COT IN MULTIMODAL VIDEO RETRIEVAL

The complete template for our reflective Chain-of-Thought (CoT) reasoning prompt designed for multimodal video retrieval is detailed in Figure 9. This structured prompt systematically integrates visual observations, personalized object attributes, and user query intentions within a unified reasoning framework. Initially, the *Original Image Description* step meticulously documents visual details from the provided reference frame, ensuring the inclusion of all relevant contextual cues. Subsequently, the *Thoughts* step explicitly interprets the user's retrieval intention by analyzing the alignment of visual attributes with personalized object usage information. The subsequent *Reflections* step involves a rigorous evaluation of identified visual and semantic elements, isolating those most congruent with the user's implicit intent. Finally, the *Target Video Description* synthesizes the reflective insights into a succinct, purpose-driven description optimized for accurate retrieval. Importantly, the reflective CoT process is encapsulated within a single comprehensive prompt, promoting coherent, efficient, and interpretable reasoning.

**Original Image Description.** In this phase, the multimodal large language model (MLLM) is tasked with comprehensively describing *all visible objects and their respective attributes (e.g., color, shape, texture, size)*. Additionally, the model must document *immediate surroundings and broader contextual factors (environmental conditions, indoor/outdoor setting)*, prioritizing precision and detail to preserve critical visual evidence essential for subsequent analytical steps.

**Thoughts.** Utilizing both the visual description and personalized object attributes (reflecting habitual usage patterns), the MLLM explicitly *interprets the retrieval intent underlying the user's query*. It identifies and elaborates on visual elements (such as dominant colors, textures, or spatial configurations) closely aligning with the user's specific object profile. Further, the MLLM incorporates semantic considerations (such as temporal sequences or action relevance) essential to accurately infer the retrieval context.

**Reflections.** In this evaluative stage, the MLLM reexamines the highlighted visual and personal object attributes from prior steps. The model critically *summarizes the integration of these visual and usage details in informing its retrieval decision*. It explicitly highlights pivotal elements (e.g., distinguishing material characteristics, contextual environment) and articulates meta-reasoning justifications to ensure coherence between the reference imagery, object attributes summary, and the user's retrieval intention. It reflects precisely on the visual or usage-derived cues that underpin its decision-making rationale.

**Target Video Description.** Utilizing refined insights from the reflective analysis, the MLLM generates a concise and targeted description pinpointing the specific video segment containing the queried object or interaction. This description is explicitly formulated as a *single, precise sentence* encompassing only retrieval-relevant elements, thus facilitating efficient and highly accurate retrieval.

### A.2.3   VISION-BY-LANGUAGE IN-CONTEXT LEARNING DETAILS

Effectively executing Reflective Chain-of-Thought (CoT) reasoning in multimodal large language models (MLLMs) requires not just general instructions but also concrete demonstrations of the reasoning process. To achieve this under a zero-shot setting without relying on direct visual guidance, we adopt a vision-by-language in-context learning (ICL) strategy inspired by recent advances in multimodal reasoning methodologies Wei et al. (2022); Mitra et al. (2024b); Zheng et al. (2023); Tang et al. (2024a).

Our Reflective CoT ICL provides MLLMs with structured language-based exemplars that guide the model through each reasoning step solely via textual information. As depicted in Figure 12, each example comprises clearly delineated components: an *Original Image Description*, *Thoughts*, *Reflections*, and a *Target Video Description*.

For clarity, consider the following example based on a user query and a provided object attributes summary:

**User Query:** *"Where was the dog after I laid the bed?"*

**Visual Reference:** *Middle frame from a reference video that shows a bed's large rectangular form with a decorative patterned cover.*

**Object Attributes Summary:** Detailed semantic attributes related to the bed, including aspects such as "decorative patterned fabric," "large size," and "used and slightly messy" status.

The Reflective CoT steps are as follows:

- **Original Image Description:** The MLLM generates a detailed depiction of visually pertinent components to the user query. In this scenario, the description captures the bed's large rectangular form with a decorative patterned cover, noting its slightly messy state indicated by creases and indentations, and contextualizes the scene within a residential bedroom with daylight filtering through partially open curtains.
- **Thoughts:** The model interprets the user's intent, identifying the dog's location post-interaction with the bed. Leveraging details from the object attributes summary (*e.g.,*

You are a highly skilled AI assistant specializing in multimodal video retrieval with deep chain-of-thought reasoning. You will receive:

Visual Reference Image
– A single frame (the middle frame) from a candidate video.
– Contains key visual details: object color, shape, texture, size, and surrounding context (indoor/outdoor, lighting, background scene).

Object Attributes Summary
– A concise personal profile of the object, including categorical and frequency data (e.g., major category, subcategory, color combinations, material, style, usage frequency).
– Reflects the user's habitual usage and personal signature.

Your task is to produce a detailed chain-of-thought explanation and a final one-sentence description of the target video. The target video is assumed to contain the object referenced in the user's query, based on both the visual evidence from the candidate frame and the user's personal usage information.

Your response must be structured as a JSON object with the following keys:
{
  "Original Image Description": <string>,
  "Thoughts": <string>,
  "Reflections": <string>,
  "Target Video Description": <string>
}

## Guidelines on Generating the Original Image Description
- Provide a thorough and detailed description of the visual reference image.
- Describe all visible elements in the reference image: the object's attributes (color, shape, texture, size), its immediate surroundings, and indoor/outdoor context.
- Be precise and comprehensive.

## Guidelines on Generating the Thoughts
- Explain your understanding of the user's query and the object attributes summary.
- Detail which visual cues (e.g., dominant colors, materials, spatial relations) align with the personal profile
- Consider semantic aspects such as Location/Positioning, Object Attributes, Temporal Sequence, Presence/Absence and Action/Manipulation.
- Discuss which details in the candidate image were most influential in guiding your decision-making process.
- Conclude with how these insights were used to formulate your final target video description.

## Guidelines on Generating the Reflections
- Summarize how the integration of the visual clues and the object attributes influenced your approach.
- Highlight the most influential details (e.g., material, environment) and why they confirm the candidate video's relevance.
- Explain how specific details (such as color, material, or setting) reinforced your decision.
- Concise meta-reasoning: justify key decisions that preserved coherence between the reference image, the attribute summary, and the retrieval goal. Highlight which visual or personal-usage cues were decisive.
- Reflect on the overall impact of these considerations in crafting a logically connected and visually coherent final description.

## Guidelines on Generating the Target Video Description
- Provide a single, concise sentence that identifies the most likely video segment containing the referenced object.

Below is an example of the expected input and output formats:
…

Figure 9: The complete template of our reflective Chain-of-Thought process for EgoMemory.

Example Input:
<Input>
{
  "User Query": "Where was the dog after I laid the bed? "
  "Visual Reference": [Attached image showing the middle frame from a video,],
  "Object Attributes Summary":
  "subcategory: bed(48)
   color: varied and patterned(11), varied(8), unspecified(7), blue(4), multicolor and patterned(4), unsure(4), blue and multicolor(3), multicolor(3), beige and green(1), blue and white(1), beige and patterned(1), white and blue(1)
   shape: rectangular(33), rectangular and slightly padded(8), rectangular and cushioned(6), rectangular and slightly cushioned(1)
   material: fabric and wood(32), wood and fabric(9), fabric and synthetic(4), wood(2), wood and synthetic(1)
   texture: soft(11), soft and slightly patterned(10), smooth and slightly soft(7), soft and slightly wrinkled(6), soft and slightly smooth(3), smooth(3), soft and slightly textured(3), soft and rough(2), smooth and slightly worn(1), soft and smooth(1), quilted(1)
   size: large(45), medium(3)
   brand: unspecified(33), unsure(15)
   style: household and resting(31), residential(15), residential and resting(1), household and sleeping(1)
   pattern: decorative and patterned(19), solid(14), decorative(7), unspecified(4), varied(3), decorative and floral(1)
   feature: supporting and sleeping(18), resting(14), comfortable and supporting(10), supporting and resting(4), comfortable and supportive(1), adjustable(1)
   usage: resting and sleeping(19), sitting(12), resting and supporting(11), resting(2), interaction and placing(1), resting and lying(1), sleeping and resting(1), residential(1)
   status: used(15), used and in use(14), used and slightly untidy(9), used and slightly messy(7), unspecified(1), used and slightly wrinkled(1), used and slightly organized(1)"
}

Example Output:
<Response>
{
  "Original Image Description": "A large rectangular bed with a decorative patterned fabric cover rests in a light-filled bedroom. The bed's surface is slightly messy, showing creases and indentations, with wooden bedframe legs visible. The surrounding room has residential furnishings, including a nightstand and partially open curtains allowing daylight.",
  "Thoughts": "The user's query focuses on the dog's location post-bed interaction. The Object Attributes Summary emphasizes the bed's 'used and slightly messy' status, 'decorative and patterned' design, and 'household and resting' context. The visual reference shows a residential bedroom setting with a disturbed bed surface, suggesting recent activity. The target video must encode both the bed's attributes (pattern, texture, usage state) and spatial context (bedroom) to align with retrieval model needs.",
  "Reflections": "The 'slightly messy' status from attributes and visible creases in the image strongly indicates recent bed interaction. The patterned fabric (mentioned in both color/texture attributes and visual description) provides distinctive visual anchors for retrieval models. Daylight through curtains in the reference image confirms the 'light-filled' environment. These elements collectively suggest the dog remained on the bed after interaction, with the patterned texture and spatial context being critical retrieval cues.",
  "Target Video Description": "A dog standing on a slightly messy, patterned bed in a light-filled bedroom."
}

Figure 10: Examples of our vision-by-language in-context learning.

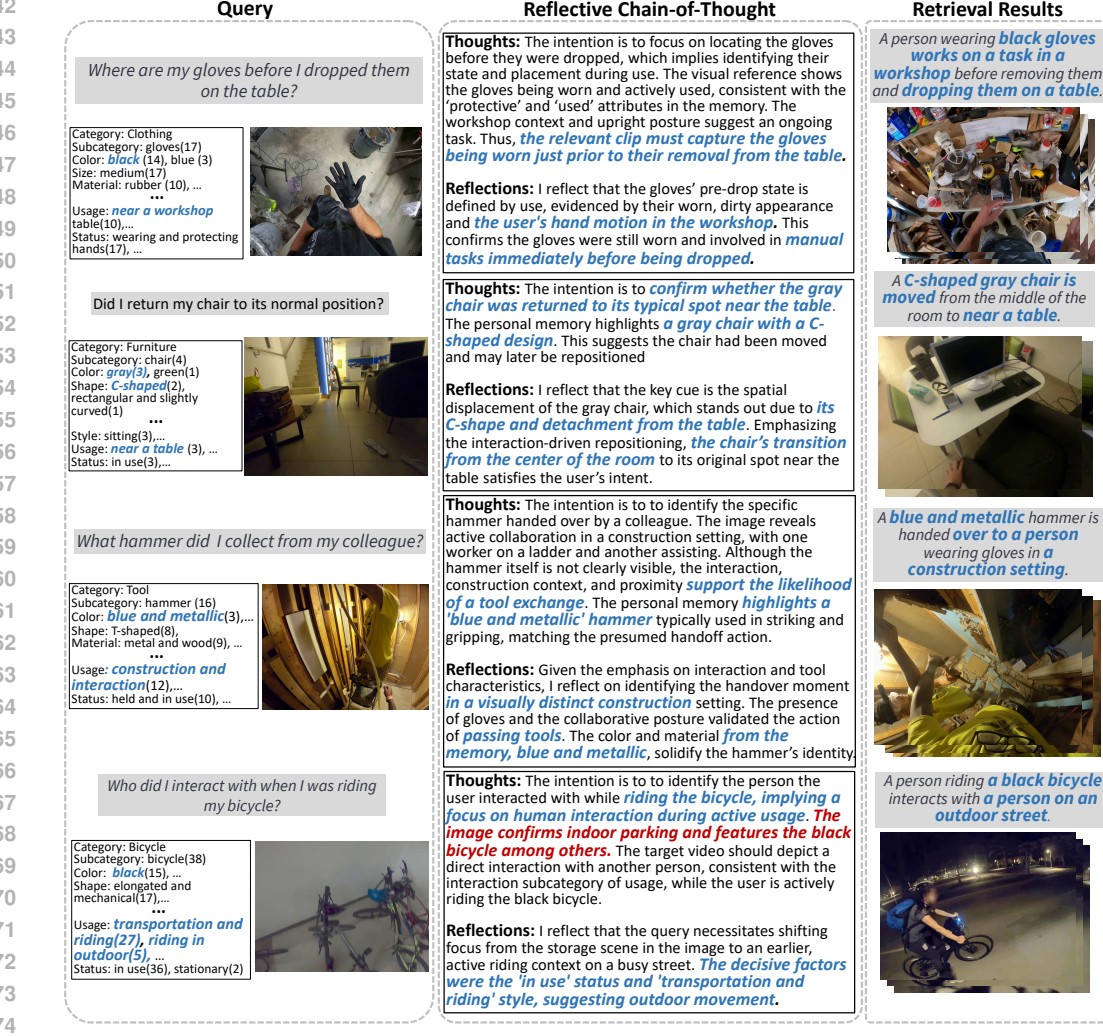

Figure 11: Qualitative analysis demonstrating the advantages of reflective Chain-of-Thought (CoT) in interpreting user intent. Row 1 illustrates how reflective CoT focuses on the relevant context of gloves being worn and used before being dropped on a table, correctly inferring their usage despite potential visual clutter. Row 4 highlights how reflective CoT discards an incorrect assumption about indoor parking and instead focuses on identifying interactions during active bicycle usage outdoors. Reflective CoT enhances the accuracy of episodic memory retrieval by filtering out irrelevant details and aligning the reasoning process with the user's true intent. Additional examples are provided in the sample data from our supplementary materials.

the bed's usage state and decorative pattern) and spatial context from the visual description (residential setting, disturbed bed surface), the model infers that the dog's presence is closely related to the bed's recent disturbance and current state.

- **Reflections:** The model explicitly reflects on its reasoning steps, evaluating how the attributes "slightly messy" and patterned fabric provide critical visual and contextual anchors for retrieval. It also notes that the daylight and residential context reinforce the recent interaction scenario, logically concluding that the dog's probable location is directly on the bed itself.

- **Target Video Description:** The model synthesizes these insights into a concise and contextually coherent description: *"A dog standing on a slightly messy, patterned bed in a light-filled bedroom."*

This structured Reflective CoT approach enables the MLLM to systematically internalize reasoning patterns from textual exemplars, supporting consistent and accurate multimodal inference even without direct visual references. By utilizing language-only in-context demonstrations, our method effectively maintains training-free adaptability, enhancing retrieval accuracy through clearly articulated reasoning pathways.

### A.2.4 MORE QUALITATIVE ANALYSIS OF REFLECTIVE CoT

To demonstrate the advantages of reflective CoT in accurately interpreting user intent, we present qualitative analyses in Figure 11. Reflective CoT plays a crucial role in filtering out irrelevant elements, such as extraneous scene descriptions or hallucinated details, which are often distractions in the reasoning process. This reflective process enhances the precision and reliability of episodic memory retrieval. In Row 1, reflective CoT excels in focusing on the relevant context by identifying the state and placement of gloves before being dropped on a table. Despite potential visual clutter, reflective CoT correctly infers the gloves' usage context, i.e., being worn and involved in manual tasks, reinforcing the memory of the gloves as protective gear. Row 4 further illustrates the power of reflective CoT by eliminating an incorrect assumption in the reasoning process. The image initially suggests indoor parking and a black bicycle, but reflective CoT shifts focus to correctly interpret the user's query about interactions during active bicycle usage outdoors. This change in focus underscores the adaptability of reflective CoT in aligning with the user's true intent, effectively disregarding irrelevant contextual details.

By integrating personal memory and contextual cues through explicit reflection, reflective CoT ensures a robust understanding of the user's intent. This not only filters out extraneous elements but also contributes to more accurate retrieval of relevant episodic memories, ultimately enhancing the system's interpretability and performance.

### A.3 DETAILS OF EGOMEMORY BENCHMARK CONSTRUCTION

To rigorously evaluate memory-augmented *personalized* egocentric video retrieval, we introduce the **EgoMemory** benchmark, constructed from Ego4D's Natural Language Queries (NLQ) Grauman et al. (2022). NLQ provides $\sim$227 hours of head-mounted video from 137 participants across 74 locations, with free-form queries about "when/where/with whom/what," closely reflecting realistic recall scenarios.

While NLQ offers rich content, its original design targets temporal localization within individual clips and does not aggregate multi-video, user-specific context. We therefore make personalization *explicit*. Concretely, we operationalize personalization as *personally experienced objects*: entities that (i) recur across a user's videos and/or (ii) are linguistically tied to the user via first-person/possessive/deictic cues. This definition does not require legal ownership; instead, it uses recurrence and linguistic evidence to avoid conflating incidental scene exposure with genuine user linkage.

To address NLQ's gaps, our benchmark treats each user as a distinct retrieval unit, aggregating all of their videos as long-term context at query time. We apply a three-stage filtering pipeline: (1) **GPT-4o CoT pre-screening** to retain queries with explicit personal references as shown in Figure 1; (2) **long-context participant selection**, requiring at least 10 videos and $\geq$1 hour cumulative footage per user to ensure sufficient temporal breadth; (3) **manual verification**, where annotators review 20 additional clips (3s each) from the same user for the queried object class and label a query "personal" if $\geq$90% of reviewed instances match ("uncertain" if $\geq$75%). This yields 639 curated queries over 45 participants (245 videos), with $\sim$91.6% labeled personal, thus emphasizing queries whose resolution benefits from cross-video personal cues rather than single-clip idiosyncrasies. Specifically:

1. **GPT-4o CoT pre-screening.** We screen all NLQ samples with a chain-of-thought prompt (As shown in the Figure) to keep queries whose main object is linguistically tied to the user (first-person possessives or deictics), while allowing secondary impersonal objects. For example, "What was the color of *my* drawstring bag?" is retained, whereas "In what aisle did I see a shopping trolley?" is excluded as a general, short-term lookup.

2. **Long context participant selection.** We define long context as users having at least 10 videos and $\geq$1 hour cumulative duration. For retained queries, the referenced main object must recur

across a user's videos, ensuring that answering the query draws on cross-video history rather than a single clip.

3. **Manual verification.** For each query–target pair, annotators inspect 20 additional 3 s clips from the same user containing the same object class. A query is "personal" if $\geq$90% of reviewed instances match the target object and "uncertain" if $\geq$75%. This process results in $\sim$91.6% personal queries and reduces conflation with scene exposure.

---

You have samples of a query and the middle frame of the target video for an **egocentric video retrieval** task, where each query may or may not represent a personalization query.
**Personalization queries** involve questions about objects or details closely related to the user's personal belongings, personal actions, or personal spaces.
Analyze the given query **step-by-step**, thinking carefully about whether it reflects a personal connection or just a general interaction or observation.

## Guidelines on Generating the **Thoughts**
1. **Identify** the object or event mentioned in the query.
2. **Consider** if the object/event typically belongs to, or is personally associated with, the user.
3. **Determine** if the query involves personal ownership, personal routine, or personal responsibility.
4. **Distinguish** between personal-related questions (about belongings, personal events, personal spaces) and non-personal general observation questions.

## Output Instructions
Based on your Thoughts, output your answer in **JSON format**, clearly indicating `"YES"` if it is a personalization query, or `"NO"` if it is not, along with your reasoning.

## Input Format
```json
<Input>
{
    "query": "<User_Query>",
    "target_video": "<Target_Video_Frame>"
}
```
## Output Format
```json
<Response>
{
    "personalization": "YES" or "NO",
    "reason": "<Your detailed reasoning>"
}
```
## Example 1
```json
<Input>
{
    "query": "What was the color of my drawstring bag?"
}
<Response>
{
    "personalization": "YES",
    "reason": "The query references a personal belonging ('my drawstring bag'), clearly indicating personal ownership."
}
```

Figure 12: The complete template of our CoT pre-screening.

To address these gaps, our benchmark treats each user as a distinct retrieval unit, aggregating all of their available videos as personal context at query time. We apply rigorous data filtering: participants

must have multiple videos, and we use GPT-4o to select queries with long-context dependencies, followed by manual curation (see Supplementary for details). The resulting dataset comprises 245 videos from 45 unique participants, spanning diverse everyday contexts and totaling 639 distinct queries.

Personalized memory banks are constructed for each participant as described in Section 3.2, resulting in 165,795 user-specific object annotations. The number of unique object types per participant ranges from 59 to 638 (median: 129), with memory bank sizes ranging from 322 to 10,454 annotations (median: 1,312). Figure 3 visualizes the most frequent objects.

The candidate retrieval set includes 2,228 video clips extracted from participants' historical egocentric videos. Each user's average candidate set size is approximately 33 clips, capturing a rich spectrum of personal and habitual contexts. Video lengths within EgoMemory span from as short as 4 seconds to a maximum of 300 seconds, averaging 103.82 seconds per clip. The total cumulative duration of the candidate set amounts to 64.25 hours. Notably, these varied temporal spans enable thorough assessments of retrieval robustness across different episodic memory lengths and complexities.

### A.4 EXPERIMENT DETAILS ON EGOCVR BENCHMARK

#### A.4.1 EGOCVR BENCHMARK OVERVIEW

To further assess the generalization ability of our proposed `EgoRetriever` framework, we conduct additional experiments on the EgoCVR benchmark Hummel et al. (2024). EgoCVR is an egocentric, composed video retrieval dataset designed to evaluate a model's capability to generate target video descriptions conditioned on reference images and textual modifications. This benchmark presents unique challenges by emphasizing nuanced temporal and semantic variations in first-person video data, complementing our primary evaluations on EgoMemory.

#### A.4.2 TASK DEFINITION.

In EgoCVR, the Composed Video Retrieval (CVR) task is defined as follows: Given a query comprising a reference frame (sampled from a video) and a free-form textual modification instruction, the objective is to retrieve the corresponding target video clip from a gallery. This retrieval is performed under two conditions: a *global* setting, where the gallery includes all test clips, and a *local* setting, where the gallery is restricted to clips from the same long-form video as the query. Formally, let $\mathcal{I}$ denote the set of reference images, $\mathcal{T}$ the set of textual modifications, and $\mathcal{V}$ the set of candidate videos. For each query $(i_q, t_q)$, the task is to identify $v^* \in \mathcal{V}$ that best reflects the semantic transformation specified by $t_q$ when applied to $i_q$.

#### A.4.3 DATASET CONSTRUCTION.

The EgoCVR dataset consists of egocentric videos capturing a diverse array of daily activities and object manipulations. Each annotated instance comprises:

- A **reference image**: a single frame extracted from the query video.
- A **textual modification**: a concise natural language instruction describing the intended change (*e.g.*, "use a different object," "perform the action faster").
- A **target video**: a short clip (2–8 seconds) from the dataset that realizes the specified modification.

Each query is paired with ground-truth target(s), and standard test splits are provided to ensure comparability across methods. For further details on dataset construction, we refer the reader to Hummel et al. (2024).

#### A.4.4 EVALUATION PROTOCOL

We follow the official evaluation protocol proposed in Hummel et al. (2024). Specifically, two retrieval settings are considered:

Table 4: Analysis the generalization of our `EgoRetriever` on the EgoCVR benchmark.

| Method | Global | | | Local | | |
|---|---|---|---|---|---|---|
| | R@1 | R@5 | R@10 | R@1 | R@2 | R@3 |
| CIReVL Karthik et al. (2024) | 2.0 | 6.8 | 10.6 | 33.6 | 49.7 | 61.4 |
| OSrCIR Tang et al. (2024a) | 4.9 | 9.3 | 13.4 | 37.4 | 53.3 | 68.1 |
| TFR-CVR Hummel et al. (2024) | 14.1 | 39.5 | 54.4 | 44.2 | 61.0 | 73.2 |
| TFR-CVR* | 14.7 | 41.2 | 55.6 | 46.1 | 62.4 | 73.9 |
| `EgoRetriever` | **17.4** | **49.2** | **62.7** | **50.3** | **68.2** | **76.4** |

- **Global Setting:** The retrieval gallery consists of all test video clips, simulating a large-scale retrieval scenario with numerous visually and semantically similar distractors.
- **Local Setting:** The retrieval gallery is restricted to clips extracted from the same long video as the reference, enabling evaluation of fine-grained, within-context retrieval.

For each query, the model generates a target description based on the reference image and textual modification, then retrieves the most relevant clip from the gallery.

### A.4.5 EVALUATION METRICS

Performance is measured using standard top-$k$ retrieval metrics:

- **Recall@K (R@$K$):** The proportion of queries for which at least one ground-truth target appears among the top $K$ retrieved results. Higher values indicate better retrieval performance.

Following the official setting Hummel et al. (2024), we report R@1, R@5, and R@10 for the global setting, and R@1, R@2, and R@3 for the local setting, following the original benchmark protocol. Metrics are averaged across all test queries.

### A.4.6 EXPERIMENTAL SETUP

All experiments are conducted on the official EgoCVR test splits, strictly adhering to the standardized evaluation settings. Model implementations and hyperparameters follow those described in Section of the main paper. The primary methods compared include:

- **CIReVL** Karthik et al. (2024): A training-free composed image retrieval approach adapted for video.
- **OSrCIR** Tang et al. (2024a): A one-stage, training-free composed retrieval framework.
- **TFR-CVR** Hummel et al. (2024): A two-stage approach utilizing video captioning and LLM-based modification.
- **TFR-CVR***: A variant using GPT-4o as both captioner and modifier.
- **`EgoRetriever`**: Our proposed one-stage, training-free framework utilizing multimodal large language models (MLLMs) and reflective chain-of-thought reasoning for precise target description generation.

### A.4.7 RESULTS

As summarized in Table 4 of the main paper, `EgoRetriever` achieves state-of-the-art performance on EgoCVR across both global and local retrieval settings. Our method consistently outperforms all baselines in Recall@K, highlighting its superior ability to generalize to novel egocentric video retrieval tasks.

### A.4.8 COMPARISON OF INPUT MODALITIES: OUR BENCHMARK VS. EGOCVR

While both our proposed benchmark (EgoMemory) and the EgoCVR benchmark Hummel et al. (2024) focus on the challenging task of composed video retrieval, a key distinction lies in the design of their input modalities and the manner in which user intent and contextual information are encoded for retrieval.

**Our Benchmark (EgoMemory):** Our evaluation protocol is motivated by real-world episodic memory retrieval, wherein the user's search intent is inherently personal and context-dependent. To capture this, we design queries to include three components:

- **Textual Memory Bank:** A collection of personal, user-centric text snippets that serve as a long-term memory repository, reflecting frequently encountered objects, past actions, or unique user experiences. This memory bank enables models to retrieve and reason over personalized historical context during retrieval.
- **Reference Image:** A visual snapshot or frame representing the starting point of the query, grounding the search in a specific visual context.
- **User Query:** A free-form natural language request, which may include references to personal context, intent, or temporal cues (*e.g.,* "Find when I last used the red mug in the kitchen").

This multimodal input setting allows models to perform retrieval that is both visually grounded and deeply personalized, supporting rich reasoning over long-context egocentric video archives.

**EgoCVR Benchmark.** By contrast, the EgoCVR benchmark Hummel et al. (2024) employs a more constrained input format, where each query consists of:

- **Reference Image:** A single frame from the query video, serving as the visual anchor for retrieval.
- **Modification Text:** A concise natural language instruction specifying a semantic change or action to be performed (*e.g.,* "pick up a different object," "use the other hand").

Notably, EgoCVR does not provide explicit access to personalized memory or long-term user context; all retrievals are based solely on the visual and textual modifications present in the immediate query. This design tests the model's ability to interpret and execute fine-grained semantic transformations but does not directly address personalization or long-horizon reasoning.

**Summary of Differences.** The primary distinction is that EgoMemory introduces a *textual memory bank* and *user query* to explicitly model personal context and long-term memory, supporting retrieval scenarios that are more representative of real-world egocentric memory augmentation. In contrast, EgoCVR focuses on visually-anchored modifications without leveraging historical or personalized information. Our experimental evaluation on both benchmarks demonstrates the generalization of our approach to settings with and without access to external memory, highlighting the flexibility and robustness of our retrieval framework.

## A.5 EXPERIMENT DETAILS ON EGOLIFEQA BENCHMARK

### A.5.1 EGOLIFEQA BENCHMARK OVERVIEW

EgoLife is a week-long, multi-person egocentric study in which six participants cohabit and continuously record with AI glasses, producing a comprehensive $\sim$300-hour multimodal dataset; EgoLifeQA is a derived QA suite tailored to long-context assistance. The released EgoLifeQA subset contains 6,000 questions over 266 hours of video.

### A.5.2 TASK DEFINITION

EgoLifeQA defines five question types: *EntityLog* (objects and their attributes/locations), *EventRecall* (past event details), *HabitInsight* (personal habit patterns), *RelationMap* (interpersonal interactions), and *TaskMaster* (task tracking and reminders). Each item is multiple-choice and is constructed to require evidence from at least five minutes prior to the question time.

### A.5.3 DATASET CONSTRUCTION

Long, "visual–audio" captions are first generated and fed to GPT-4o to propose timestamped QA candidates; annotators then filter and refine them, retaining only questions with long-range dependencies (at least five minutes) and high real-world relevance, yielding the final QA set.

### A.5.4 EVALUATION PROTOCOL

We follow the official setting: models must answer the fixed-choice questions using only the benchmark inputs and produce supporting predictions per category. :contentReference[oaicite:7]index=7

### A.5.5 EVALUATION METRICS

Performance is reported as accuracy (%) per question type, averaged over the official test questions (EntityLog, EventRecall, HabitInsight, plus the remaining categories as applicable).

### A.5.6 EXPERIMENTAL SETUP

Implementations and hyperparameters mirror those in the main paper. We compare `EgoRetriever` to **GPT-4o**, **LLaVA-OV**, and **EgoGPT** under identical inputs; `EgoRetriever` uses the same MLLM backbone (GPT-4o) and our reflective reasoning to form answers. :contentReference[oaicite:9]index=9

### A.5.7 RESULTS

As summarized in Table 4 `EgoRetriever` attains the highest accuracy on **EntityLog**, **EventRecall**, and **HabitInsight** (42.5, 45.1, 37.7), outperforming the strongest baseline by +3.3, +3.0, and +6.6 points, respectively, demonstrating transfer beyond retrieval to fixed-answer, long-context QA focused on entities, events, and habits.

### A.6 MORE ABLATION STUDY

Table 5: Additional ablation results in terms of mR@1, mR@2, and mR@3, demonstrating the necessity of one-stage reasoning and the superiority of our Reflective CoT design.

| | Methods | mR@1 | mR@2 | mR@3 |
|---|---|---|---|---|
| 1. | Full Model (Reflective CoT, one-stage, middle reference frame) | 23.19 | 38.48 | 47.83 |
| **Significance of the one-stage reasoning strategy** | | | | |
| 2. | Two-stage | 18.21 | 27.12 | 32.05 |
| 3. | Two-stage + CoT | 19.92 | 32.41 | 40.73 |
| **Alternative prompting strategies for Reflective CoT** | | | | |
| 4. | Simple prompt | 19.27 | 31.48 | 40.11 |
| 5. | Simple CoT (no reflection and designed thoughts) | 20.13 | 33.18 | 41.28 |
| 6. | DDCoT Zheng et al. (2023) | 19.42 | 32.50 | 41.02 |
| **Alternative reference frame select strategies for Reflective CoT** | | | | |
| 7. | Random selected | 23.04 | 38.02 | 47.16 |
| 8. | Context-based selected | 23.39 | 38.94 | 48.11 |
| **Dependence on human narrations** | | | | |
| 9. | 25% narrations dropped | 22.46 | 36.21 | 46.09 |
| 10. | 50% narrations dropped | 21.82 | 35.19 | 45.32 |

Table 5 presents further ablation results that complement the analysis in the main paper by evaluating the significance of our one-stage reasoning pipeline, the role of different CoT variants, and reference frame selection strategies.

**(1) Models '2–3' assess the significance of the one-stage reasoning strategy.** We compare our unified reflective inference process with a two-stage pipeline where the reference caption and final description are generated sequentially. The two-stage baseline (model '2') yields an mR3 of 32.05, which is 15.78% lower than the full one-stage model (model '1'). Incorporating basic Chain-of-Thought prompting into the two-stage pipeline (model '3') improves performance to 40.73, yet still underperforms our one-stage reflective reasoning, indicating the effectiveness of jointly reasoning over query, reference, and memory in a single coherent pass.**(2) Models '4–6' explore alternative prompting strategies for reflective reasoning.** Replacing our Reflective CoT with a simple, flat instruction (model '4') or a simple CoT strategy (model '5') leads to 6.21% and 4.97% average performance drops, respectively, confirming the necessity of deep multimodal reasoning. Further, we evaluate a structured CoT variant using DDCoT Zheng et al. (2023), which first decomposes user queries before composing descriptions (model '6'), resulting in the lowest performance among CoT variants. This suggests that sequential decomposition may hinder holistic interpretation in personalized video contexts, reinforcing the advantage of our proposed Reflective CoT.**(3) Models '7–8'**

Table 6: Comparative analysis of computational cost, latency, memory usage, API expenditure, and retrieval performance across baseline and proposed models.

| Model | LLM | Latency | GPU Memory | API Cost | Avg. Performance |
|---|---|---|---|---|---|
| CIReVL | GPT-3.5 | $\sim 1.4$s | 40 GB | $\sim \$0.001$ | 24.86 |
| OSrCIR | GPT-4o | $\sim 0.7 \pm 0.08$s | 16 GB | $\sim \$0.004$ | 27.87 |
| TFR-CVR | GPT-4o | $\sim 1$s | 16 GB | $\sim \$0.007$ | 41.25 |
| EgoRetriever | GPT-4o-mini | $\sim 0.5 \pm 0.05$s | 16 GB | $\sim \$0.002$ | 46.92 |
| EgoRetriever | GPT-4o | $\sim 0.7 \pm 0.08$s | 16 GB | $\sim \$0.004$ | 48.19 |

**evaluate alternative reference frame selection strategies.** Compared to the randomly selected frame baseline (model '7'), our middle-frame selection strategy (model '1') provides a slightly improved retrieval performance (0.427% higher on average). An alternative frame selection method, which leverages clip embeddings to calculate the similarity between captions and individual frames (model '8'), achieves marginally higher performance (0.313% higher on average). However, this embedding-based approach significantly reduces efficiency due to the computational overhead of evaluating similarity against dense frame-level descriptions provided by Ego4D's narrations (at 30 fps). Therefore, our simpler middle-frame strategy is optimal, striking an effective balance between retrieval accuracy and computational efficiency. **(4) Model '9-10' analysis the dependence on human narrations of our personal memory bank.** Dropping 25% (model '9') and 50% (model '10') of human narrations yields moderate degradations, consistent with redundancy in our memory bank: recurring personal objects are recorded across clips and frequency-based filtering suppresses noise from missing narrations.

### A.7 DETAILED COMPARISON OF COMPUTATIONAL COST

In Table 6 we report a four-way cost–effectiveness analysis, mean query latency (averaged over 100 runs), peak GPU memory at inference, OpenAI API expenditure per call, and average retrieval accuracy (ViT-L/14 backbone) on EgoMemory and EgoCVR, covering two prior baselines (CIReVL and OSrCIR), the two-stage TFR-CVR, and our single-stage EgoRetriever variants. CIReVL, which relies on a GPT-3.5 captioning head plus a separate retrieval LLM, incurs the highest memory footprint (40 GB) and the slowest response time ($\sim$1.4 s), despite its modest API fee, and delivers the lowest accuracy (24.86). OSrCIR and TFR-CVR both reduce memory to 16 GB by avoiding a BLIP-2 captioner; however, OSrCIR's one-stage design halves latency to $\sim$0.7 s but still trails TFR-CVR by 13.4 mR points. Our EgoRetriever with GPT-4o-mini preserves OSrCIR's low memory/latency profile while lifting performance to 46.92, and the full GPT-4o variant maintains this latency ($\sim 0.7 \pm 0.08s$) while achieving the highest accuracy (48.19) without increasing memory consumption. Taken together, these results show that (i) eliminating separate captioning modules yields substantial memory and speed benefits, (ii) mini-scale MLLMs already strike an excellent cost–performance balance, and (iii) our reflective one-stage pipeline offers the best absolute accuracy with competitive operational costs, underscoring its suitability for real-time, resource-constrained egocentric retrieval systems.

### A.8 SCALABILITY UNDER INCREASING VIDEO/METADATA VOLUME

Table 7: Scalability with dataset size. Latency is averaged over 100 queries; API cost is per call.

| # Videos | Avg. Latency (s) | GPU Memory (GB) | API Cost (USD) |
|---|---|---|---|
| 10 | $0.65 \pm 0.03$ | 16 | $\sim \$0.004$ |
| 50 | $0.70 \pm 0.05$ | 16 | $\sim \$0.004$ |
| 100 | $0.70 \pm 0.07$ | 16 | $\sim \$0.004$ |
| 245 | $0.70 \pm 0.08$ | 16 | $\sim \$0.004$ |

**Findings.** As Table 7 shows, query-time latency remains $\approx$0.65–0.70 s with small variance as the corpus grows from 10 to 245 videos; peak GPU memory is flat at 16 GB and per-query API expenditure is unchanged. This stability stems from EgoRetriever using precomputed video and memory embeddings, so online retrieval reduces to lightweight lookups and similarity scoring; only

offline indexing scales with data volume and does not affect interactive latency. In tandem with Fig. 6 (main), where larger memory-bank context improves accuracy, these results indicate that `EgoRetriever` scales efficiently in both cost and effectiveness as personalized data accumulates.

## A.9 CANDIDATE SET STATISTICS

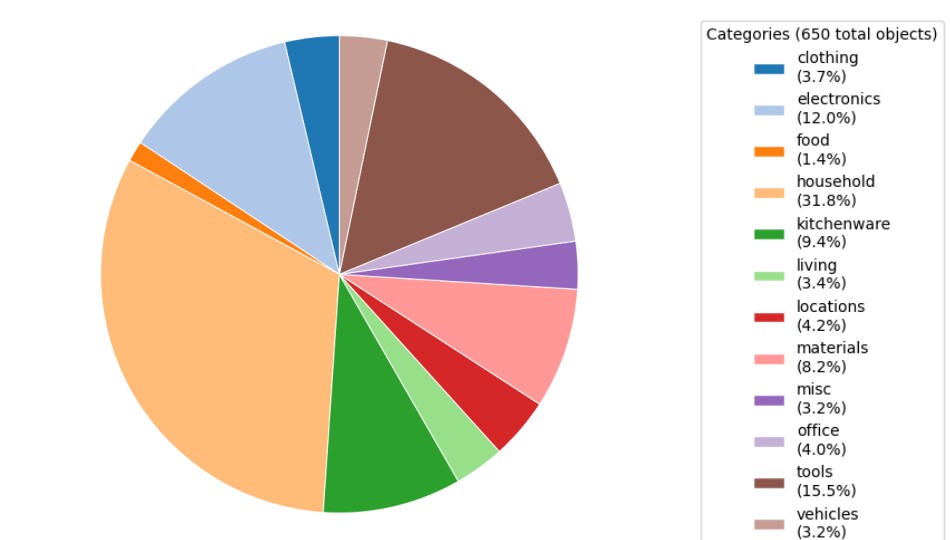

Figure 13: Distribution of 650 object instances across 13 semantic categories in the candidate retrieval set. The largest categories are household (31.8%), tools (15.5%), and electronics (12.0%), reflecting the diversity and realism of egocentric video environments.

To facilitate personalized episodic memory retrieval, we construct a candidate retrieval set for each participant by aggregating their historical egocentric video clips prior to each query. This design simulates realistic personal memory banks and enables a thorough evaluation of retrieval models across diverse user contexts.

**Composition and Scale.** The benchmark candidate set comprises a total of 2,228 video clips collected from 45 unique users, sampled from their continuous head-mounted video recordings. The number of candidate clips per participant ranges from 9 to 61, with an average of 33 clips per user. Clip durations vary from 4 to 300 seconds, with a mean duration of 103.82 seconds, resulting in a cumulative total of 64.25 hours of video data in the candidate pool.

**Object Instance Distribution.** Within the candidate set, we annotated 650 object instances spanning 13 semantic categories. Figure 13 illustrates the distribution of these object categories. The largest proportions are household objects (31.8%), tools (15.5%), and electronics (12.0%), alongside kitchenware (9.4%), materials (8.2%), and other everyday object classes. This distribution reflects the diversity and complexity of real-world personal environments encountered in egocentric video.

**Personalization and Diversity.** To assess diversity, we computed Jaccard similarity scores for the top 100 most frequent object types based on their attribute metadata. Inter-participant similarity was below 0.4 for 74% of object types, indicating high heterogeneity between users. Furthermore, intra-participant similarity was consistently higher than inter-participant similarity, underscoring the personalized and user-specific nature of each candidate set.

**Evaluation Metric.** For all experiments, we report **mean Recall@K** (K=1, 2, 3), computed separately for each user over their candidate set and then macro-averaged across users. This approach, consistent with prior benchmarks Grauman et al. (2022); Hummel et al. (2024), ensures that performance reflects the retrieval difficulty for each individual and mitigates biases due to uneven query counts. In cases with a single correct answer per query, Recall@K is equivalent to Hit Rate@K Sun et al. (2019), a standard metric in recommender systems.

## A.10 MORE IMPLEMENTATION DETAILS

### A.10.1 DETAILS OF RIGOROUS DATA FILTERING

To ensure rigorous evaluation and the authenticity of personalized retrieval contexts in the **EgoMemory** benchmark, we implemented a comprehensive data filtering process. Initially, we screened the Ego4D Natural Language Queries (NLQ) dataset Grauman et al. (2022), restricting inclusion criteria to participants possessing multiple egocentric videos. This criterion was essential to authentically simulate realistic retrieval scenarios, as genuine personal retrieval contexts inherently involve multiple video interactions over extended periods.

Next, we utilized GPT-4o to systematically identify and select queries exhibiting clear long-context dependencies. Specifically, GPT-4o was prompted to assess each query's dependency on temporal context beyond immediate video boundaries, prioritizing queries whose interpretations or resolutions necessitate referencing historical, user-specific contexts. Examples of selected queries typically involved repeated interactions, habitual activities, or persistent object engagements spanning multiple video sessions.

Following the AI-driven initial screening, we conducted meticulous manual curation to verify query suitability rigorously. Expert annotators reviewed each GPT-4o-identified query, confirming genuine long-context relevance and filtering out queries ambiguous in contextual dependence or insufficiently representative of personalized retrieval demands. Critically, filtered samples required that participants have various candidate video clips, specifically more than 10 videos per participant, with an average candidate set size of approximately 33 clips. This ensured a robust and diverse long-context retrieval environment, capturing rich habitual and episodic nuances.

The resulting refined dataset comprises 245 rigorously filtered videos from 45 participants, providing 639 distinct, carefully validated long-context queries. This comprehensive filtering approach significantly enhances the benchmark's effectiveness in accurately assessing personalized retrieval capabilities under authentic, user-specific episodic memory contexts.

### A.10.2 EXTENDED IMPLEMENTATION DETAILS FOR EGOMEMORY AND EGORETRIEVER

We leveraged GPT-4o extensively to construct detailed user-specific memory banks by generating comprehensive object-centric metadata annotations from video clips. Additionally, GPT-4o facilitated reflective Chain-of-Thought (CoT) reasoning integral to our retrieval framework. Specifically, we conducted these annotation and reasoning processes for 45 participants, covering a total of 245 videos, over approximately one month. This substantial annotation effort underscores both the computational and temporal investments involved in establishing robust personalized memory contexts.

Our retrieval experiments utilized four NVIDIA V100 GPUs, each equipped with 32GB memory. We evaluated multiple state-of-the-art video-language models, including LanguageBind Zhu et al. (2023), CLIP Radford et al. (2021), BLIP Li et al. (2022), and EgoVLPv2 Pramanick et al. (2023). Within our proposed `EgoRetriever`, EgoVLPv2 served as the primary text encoder. To represent videos visually, embeddings from CLIP and BLIP were computed by averaging across embeddings extracted from 15 uniformly sampled frames per video clip. For each candidate video, these visual embeddings were then matched with the GPT-4o-generated textual descriptions using cosine similarity to determine retrieval accuracy. For further detailed methodology and specific parameter settings, please refer to the supplementary materials.

## A.11 LIMITATIONS

Although `EgoRetriever` demonstrates strong performance and significant advances in personalized long-context egocentric video retrieval, several limitations warrant consideration. First, our model critically depends on accurate extraction and annotation of user-specific object-centric metadata via Multimodal Large Language Models (MLLMs), implicitly assuming these annotations to be precise and exhaustive; inaccuracies or omissions could substantially degrade retrieval effectiveness. Second, as highlighted by our failure analysis, the framework frequently encounters difficulties in object disambiguation within cluttered scenes and struggles with context interpretation when visual cues are ambiguous or insufficient, indicating sensitivity to visual clarity and query specificity. Third, the Ego4D dataset, from which our benchmarks are constructed, lacks explicit

temporal annotations across different videos for individual participants, limiting the capability to accurately reconstruct chronological timelines of user experiences, which is essential for genuinely long-context episodic retrieval. Furthermore, while our evaluation validates generalizability using the EgoMemory and EgoCVR benchmarks derived from Ego4D, the method has not yet been extensively assessed across other diverse egocentric datasets, potentially constraining broader applicability. Finally, the reliance on computationally intensive MLLMs and extensive metadata annotations raises scalability and efficiency concerns, particularly for deployment on resource-limited wearable devices, emphasizing the necessity of future research into streamlined model optimization and annotation methodologies.

## A.12 BROADER IMPACTS

Our proposed framework, `EgoRetriever`, introduces significant advancements toward personalized, long-context episodic memory retrieval, opening promising avenues for augmented human cognition and improved assistive technologies. However, its deployment also brings potential societal risks warranting careful consideration. First, extensive recording and storage of personalized egocentric data inherently pose substantial privacy concerns, as such continuous visual and contextual capture could inadvertently reveal sensitive personal information or be exploited for unauthorized surveillance and tracking. Secondly, even when functioning correctly, the model might unintentionally reinforce biases embedded in training data or annotations, possibly leading to unequal performance across diverse demographic groups, thus raising fairness considerations. Additionally, misuse of the proposed memory-augmented retrieval technology could facilitate invasive monitoring or targeted manipulation based on personal habits and behaviors, resulting in malicious outcomes such as stalking, identity theft, or psychological manipulation. Incorrect retrieval outcomes, particularly involving sensitive contexts or critical decisions, could also lead to harmful personal or societal consequences. To mitigate these risks, we advocate implementing robust privacy-preserving measures, including data encryption, strict access controls, and user-centric data ownership frameworks. Moreover, comprehensive fairness audits and continuous model evaluations across diverse user populations are essential to ensure equitable deployment. Finally, careful consideration of transparent usage policies and developing detection mechanisms for identifying and preventing misuse are crucial steps toward responsibly harnessing the full potential of personalized egocentric video retrieval technologies.

## A.13 EXTENDED RELATED WORKS

**Multimodal Chain-of-Thought Reasoning for Egocentric Video Retrieval.** Multimodal Large Language Models (MLLMs) have recently demonstrated remarkable reasoning capabilities, particularly when equipped with Chain-of-Thought (CoT) prompting strategies Wei et al. (2022); Kojima et al. (2022); Zheng et al. (2023); Zhang et al. (2023). In the domain of composed image retrieval (CIR), Recent advances introduced OSrCIR Tang et al. (2024a), a training-free, one-stage reflective CoT framework that enables MLLMs to reason about manipulation intent and preserve contextual information, thereby improving retrieval accuracy without the typical information loss associated with two-stage approaches. Similarly, EmbodiedGPT Mu et al. (2023) extends CoT prompting to vision-language pre-training in embodied environments, leveraging an EgoCOT dataset and prefix-tuned LLMs to enable agents to perform complex action planning through multimodal reasoning. Despite these advances, existing research on episodic memory retrieval from egocentric videos has predominantly focused on short-term or single-video scenarios Grauman et al. (2022); Hummel et al. (2024); Yang et al. (2025), with limited attention to the long-context, personalized nature intrinsic to human memory. Current benchmarks lack comprehensive personal memory banks and rich, user-specific object annotations Singh et al. (2016); Damen et al. (2020); Núñez-Marcos et al. (2022); Wang et al. (2023b), which are critical for modeling real-world episodic recall. Building upon recent advances in multimodal CoT reasoning Tang et al. (2024a); Mu et al. (2023); Mitra et al. (2024a); Zheng et al. (2023); Zhang et al. (2024), our work is the first to bring this paradigm to personalized egocentric video retrieval. We introduce `EgoRetriever`, a training-free framework that combines MLLMs with reflective CoT prompting, leveraging a comprehensive personal memory bank constructed from extensive user-specific object annotations in Ego4D Grauman et al. (2022). This enables the model to interpret nuanced user queries and generate detailed, contextually grounded descriptions of target video clips by incorporating recurring personal objects, habitual activities, and social interactions. Extensive experiments on the EgoMemory and EgoCVR Hummel

et al. (2024) benchmarks demonstrate that `EgoRetriever` consistently outperforms existing baselines, underlining its strong generalizability and promise for real-world deployment in personalized episodic memory retrieval.

### A.14 THE USE OF LARGE LANGUAGE MODELS (LLMS)

LLMs were used in two distinct ways: as core research components of our method and as general–purpose writing aids. The authors take full responsibility for all content; LLMs are not authors.

**Role in research methodology.** We use multimodal LLMs strictly at inference time, without fine–tuning, as part of our proposed system: (i) GPT-4o and GPT-4o-mini are used for reflective chain-of-thought reasoning to generate target video descriptions, (ii) GPT-4o is used for CoT pre-screening of NLQ queries during dataset filtering, and (iii) LLaVA and Qwen2.5-VL serve as open-source alternatives in ablations. Prompts, pipelines, and hyperparameters are provided in the Appendix for reproducibility. When hosted APIs were used, we transmitted only minimal, structured metadata (and reference frames where explicitly stated), never raw egocentric streams.

**Role in writing.** LLMs (GPT-4 class) were used to assist with copy-editing, grammar, and wording suggestions (e.g., tightening paragraphs, standardizing terminology, improving figure/table captions). They did not originate scientific claims, experimental designs, or citations; all technical content, analyses, and references were authored, verified, and curated by the authors.

**Accountability and authorship.** All results, claims, and text are the responsibility of the authors. We checked model outputs for accuracy and potential plagiarism. LLMs are not eligible for authorship and are credited here solely as tools.

**Data handling.** We avoided sharing personally identifying content with hosted models; no user identities, faces, audio, GPS traces, or full-length videos were transmitted. Open-source MLLMs enable fully local execution when stricter privacy is required.

**Reproducibility.** We document model versions, prompts, and settings (Appendix) and note the inherent non-determinism of API LLMs. Any deviations are reported in the experimental details.

