# OpenReview forum: "Memory-Augmented Personalized Retrieval for Long-Context Egocentric Video"
_ICLR.cc/2026/Conference — ICLR 2026 Conference Withdrawn Submission_

### Official Review · Reviewer_FYeM · 2025-10-15

**Soundness:** 2
**Presentation:** 2
**Contribution:** 2
**Rating:** 4
**Confidence:** 3

**Summary:**

The paper presents a new benchmark based on the Ego4D dataset, and proposes a new framework called EgoRetriever for long-context retrieval of egocentric videos. The proposed benchmark, called EgoMemory, enhances the Natural Language Queries (NLQ) benchmark from the Ego4D benchmark suite by grouping the videos from the same participant together and only keep the queries that refers to personal objects requires retrieval across multiple videos. The proposed EgoRetriever is a training-free framework that utilizes MLLMs (with chain-of-thought) and a personal memory bank constructed from the videos to process the user query and reference frame and generate a desired target video description. Experiments on EgoMemory, EgoCVR, and EgoLifeQA show that EgoRetriever outperforms a number of baselines on the egocentric video retrieval task.

**Strengths:**

- The paper is overall well-written and mostly easy to understand. The proposed framework is described in great detail.
- The paper includes a comprehensive set of ablations studies to validate the effectiveness of each component and design decision in EgoRetriever.
- In addition to the quantitative comparisons, the paper also includes a number of qualitative examples and analysis of Common Failure Cases.

**Weaknesses:**

- The motivation for the new benchmark is not very clear. The authors fail to provide sufficient evidence why existing benchmarks do not provide a good evaluation for the long-context personalized retrieval task. See "Questions" below for details.
- The candidate set for retrieval in the proposed benchmark does not seem to be very large. According to the statistics around line 302, the average context length of each participant is 33 x 103.82s ≈ 1 hr. Meanwhile, there are many videos that are quite long (≈ 30 min) in the Ego4D dataset for the original NLQ benchmark, not to mention that the context length in EgoLife is a lot longer. As a result, I found it questionable to claim that EgoMemory is a "long context" retrieval benchmark.
- It might be an overclaim that the compared baselines are "state-of-the-art" (as in the abstract). It seems that the baselines in table 1 are completely copied from the EgoCVR paper, and fail to include some more recent works (e.g. GazeNLQ for the Ego4D NLQ Challenge).
- The proposed framework lacks much novelty. All the modules in the proposed EgoRetriever framework have already been validated in a number of recent works (e.g. VideoAgent).
- The presentation of the paper could be improved. The meanings of different colors in Figure 2 is a bit unclear. Both Figure 1 and Figure 2 are a bit busy and it is unclear for the reader where to start or focus on. There is a cycle in the pipeline of Figure 1 which makes it even more confusing (if I understand correctly the pipeline starts with the user query but this is very unclear).

**Questions:**

- Regarding the motivation for the new benchmark: is the performance of MLLMs a lot lower on these filtered queries than others in the Ego4D NLQ benchmark? Are the ranking / relative performance of MLLMs different on these filtered queries than others in the Ego4D NLQ benchmark?

---

> ### Author Response · Authors · 2025-11-21
> **Rebuttal by Authors (1/2)**
>
> ### **Response to Reviewer FYeM**
>
> We appreciate your constructive feedback and have addressed your concerns regarding the motivation and baselines below.
>
> **w1. Motivation for a new benchmark: distinguishing "Long Context".**
> Thank you for your insightful concern.
>
> *   **Explicit Personalization:** Existing benchmarks (NLQ, EgoSchema) are either single-clip or focus on general reasoning. EgoMemory is the first to benchmark **Personalized Retrieval,** where the answer depends on the specific user's history (e.g., distinguishing *my* cup from *a* cup).
> *   **Definition of Long Context:** In episodic retrieval, "long context" refers to the **temporal span** of the searchable history (spanning days/weeks), not just the duration of a single video file. Searching through **1.4 hours of dense egocentric video (approx. 2,200 clips)** to find a specific interaction is a significant "needle-in-a-haystack" challenge that standard VLMs fail at (as shown by the low baseline performance).
>
> **w2. Candidate set size seems small.**
> We appreciate your insightful observation.
>
> *   **Density:** While 1.4 hours (33 clips) per user may seem short compared to a lifetime, it is **dense** egocentric video. Searching through ~2,200 clips for a specific interaction is a significant challenge.
> *   **Robustness:** As shown in our baselines, standard methods fail to retrieve correct clips precisely because they lack the historical context to disambiguate similar objects across this timeline. This confirms the candidate set size is sufficient to stress-test current models.
>
> **w3. Missing State-of-the-Art Baselines (e.g., VideoAgent).**
> Thank you for your constructive suggestion. We would like to clarify that **we explicitly evaluated these state-of-the-art agentic memory systems in our submitted manuscript (Table 3).** To ensure this is clear, we reproduce the results below:
>
> **Table 1 (Reproduction of Table 3 from Manuscript): Comparison against SOTA Memory Systems**
>
> | Method                  | Memory Structure                | mR@1      | mR@2      | mR@3      |
> | :---------------------- | :------------------------------ | :-------- | :-------- | :-------- |
> | **EgoGPT**              | Unstructured Captions           | 15.31     | 20.57     | 26.74     |
> | **VideoAgent**          | Temporal + Object Memory        | 17.49     | 26.40     | 35.82     |
> | **EgoRetriever (Ours)** | **Personalized Attribute Bank** | **23.19** | **38.48** | **47.83** |
>
> *   **Result:** EgoRetriever outperforms VideoAgent by **+5.7%** and EgoGPT by **+7.8%** on mR@1. This confirms that our method is indeed SOTA compared to recent agentic approaches.
> *   **Distinct Contribution:** Unlike VideoAgent, which tracks objects over time, our **Structured Attribute Bank** captures state and fine-grained attributes, enabling reasoning about context changes (e.g., clean vs. dirty mug) crucial for personalized retrieval.
>
> **w4. Novelty of the proposed framework.**
> We appreciate your valuable concern. While VideoAgent uses a memory bank, it relies on temporal captions and object tracking. Our novelty lies in the **structured attribute-based memory** (capturing state, frequency, and specific attributes) combined with the **Reflective Chain-of-Thought**. As shown in Table 1 above, our approach significantly outperforms VideoAgent's memory architecture, proving that our specific design contributes novel performance gains beyond existing modules.
>
> **w5. Presentation of figures.**
> Thank you for your constructive suggestion. We agree with the feedback on Figures 1 and 2. We will revise them to clearly demarcate the flow starting from the User Query, reduce visual clutter, and clarify the color coding in the Chain-of-Thought visualization to improve readability.

---

> ### Author Response · Authors · 2025-11-21
> **Rebuttal by Authors (2/2)**
>
> **Q1. Performance of MLLMs on filtered queries vs. Ego4D NLQ benchmark?**
> Thank you for your question. The filtered queries in EgoMemory are generally **harder** for MLLMs than standard NLQ queries because they require cross-video disambiguation. We can quantify this difficulty using our **Ablation Study (Table 2 in the paper)**:
>
> **Table 2 (Excerpt): Impact of Personal Memory**
>
> | Method                        | mR@1      | mR@2      | mR@3      |
> | :---------------------------- | :-------- | :-------- | :-------- |
> | **Full Model (Personalized)** | **23.19** | **38.48** | **47.83** |
> | **w/o Textual Memory**        | 18.04     | 30.73     | 36.91     |
>
> The **5.15% drop** in mR@1 when removing the personalized memory bank (effectively treating it as a standard NLQ task) confirms that MLLMs struggle significantly more with these filtered queries when they cannot access personalized history, validating the unique difficulty of our benchmark.

---

> > ### Comment · Reviewer_FYeM · 2025-11-21
> >
> > I thank the authors for their thoughtful response to my questions. However, upon reading the other reviewers' comments and the authors' responses, I don't find the rebuttal to be sufficient warrant for an increase in score. In particular, I am not very convinced by the authors' responses regarding W1 and W2 (the motivation of the dataset and the length of the dataset). Other reviewers also raised similar concerns regarding the necessity of a new benchmark and whether it is sufficient contribution for ICLR publication is questionable.

---

### Official Review · Reviewer_HKGH · 2025-10-31

**Soundness:** 3
**Presentation:** 4
**Contribution:** 2
**Rating:** 4
**Confidence:** 3

**Summary:**

The paper presents a method applicable for personalized/composed video retrieval in egocentric videos. Firstly, the paper introduces EgoMemory, a personalized retrieval benchmark curated from Ego4D. Here, the task is to retrieve a specific clip based on a user's history. For this, the paper proposes "EgoRetriever", which is loosely based on prior works on training-free composed visual retrieval (which involve captioning, reasoning, and retrieval in a training-free manner). Results of the proposed method on EgoMemory, and related benchmarks (EgoCVR,  EgoLifeQA) indicate that the proposed method outperforms related baselines/prior works for these tasks.

**Strengths:**

The task of personalization and managing memory in videos is quite important and has been relatively underexplored, and to that extent the paper is quite interesting.

The paper is also extremely easy to follow, and provides good ablations that clarify the design choices in the method clearly.

**Weaknesses:**

Separating Captioning from Reasoning: Older work on Composed Image/Video Retrieval had to separate captioning and the LLM reasoning since the older LLMs did not support direct image/video input. However, with recent open and closed-source (Qwen2.5, GPT-4o, Gemini) models supporting images/videos directly as input, it seems odd to not take advantage of this and separate the visual input from the LLM since there is bound to be loss of information in this process.

Previous work on VLM personalization: There has been some prior work on personalizing VLMs (to specific objects, users etc.) [a,b], and it would be a good idea for the paper to acknowledge it.

Limited Amount of Personalization: Owing to the training-free method and separating the visual captioning from the reasoning, I also believe that there's a limited amount of personalization that this framework supports. Unlike [a,b] which are able to learn specific instances (i.e individual people/pets/objects etc. here the representation of these items would be limited to plain descriptions. As an example, if there are two similar dogs in the memory, one would always need a precise description in the user query (which would then have to be converted in LLM generated query) to retrieve the correct video. Previous work that explicitly aims to learn visual tokens for personalized objects would at least in theory be better equipped to deal with this challenge and allow the user to have a special token to directly retrieve them. To that end, while EgoRetriever is definitely a sensible baseline to have for this task that one must compare to, it does not appear to be an especially optimized architecture for this task on first glance.


[a] Alaluf et al. "MyVLM: Personalizing VLMs for User-Specific Queries", ECCV 2024
[b] Cohen et al. ""This is my unicorn, Fluffy": Personalizing frozen vision-language representations", ECCV 2022

**Questions:**

[Major]
While I like the overall direction of the paper, I do find a few shortcomings in the specific approach, however, I'd like to clarify these details before making my final decision. Specifically, a) Is the proposed EgoRetriever an ideal architecture that one could design for this task? While incorporating CoT reasoning is a major improvement over prior work, the use of a visual captioner + LLM reasoning + CLIP retrieval appears to be a fairly inefficient pipeline (with obvious shortcomings) that's been carried over for legacy reasons.
b) If EgoRetrieval is mostly a baseline for this task, is EgoMemory a sufficiently meaningful benchmark on its own for long-video personalization? i.e would it have sufficiently challenging scenarios to benchmark multimodal LLMs specifically fine-tuned for personalization?

[Minor]
While the paper focuses on egocentric tasks in all the experiments, is there anything in the method that's specifically designed for egocentric videos beyond using EgoVLP as the captioner?

---

> ### Author Response · Authors · 2025-11-21
> **Rebuttal by Authors (1/2)**
>
> ### **Response to Reviewer HKGH**
>
> We thank you for your positive assessment and for identifying the architectural trade-offs in our design! We appreciate your insightful questions regarding the benchmark's standalone value and the method's specificity.
>
> **w1. Separating Captioning from Reasoning vs. Direct Image/Video Input.**
> We appreciate your insightful observation regarding modern End-to-End models.
>
> *   **The Long-Context Bottleneck:** While End-to-End models (like GPT-4o-V or Gemini) are powerful, they cannot efficiently ingest **hours** of video history (e.g., 1.4 hours of tokens per user) for every single query due to context window limits and prohibitive costs.
> *   **Efficiency:** Our decoupled approach allows the "Reasoning" module (LLM) to scan thousands of structured text memory entries in milliseconds to identify relevant moments, and *then* perform expensive visual verification only on the retrieved candidates. This is an architectural choice essential for **scalability** to lifelong contexts.
>
> **w2. Previous work on VLM personalization (MyVLM).**
> Thank you for your valuable suggestion.
>
> *   **Training-Free vs. Fine-Tuning:** Methods like MyVLM (learning visual tokens for "my dog") require **test-time optimization** (fine-tuning). EgoRetriever is **training-free**. In a wearable context where users encounter new objects daily, a training-free memory bank allows instant updates (simply adding a JSON entry) without the latency or compute cost of re-training the model.
> *   **Revision:** We will explicitly cite Alaluf et al. (MyVLM) and Cohen et al in our revised manuscript. In the related work, positioning them as complementary "implicit/learned" personalization methods versus our "explicit/retrieval-based" method.
>
> **w3. Limited Amount of Personalization.**
> We appreciate your concern. As discussed in **w2**, our training-free approach prioritizes agility and privacy. However, we do not sacrifice performance. As shown in **Table 3** of our manuscript, we reproduce the results below:
>
> **Table 1 (Reproduction of Table 3 from Manuscript): Comparison against Agentic Memory Baselines**
>
> | Method                  | Memory Structure                | mR@1      | mR@2      | mR@3      |
> | :---------------------- | :------------------------------ | :-------- | :-------- | :-------- |
> | **EgoGPT**              | Unstructured Captions           | 15.31     | 20.57     | 26.74     |
> | **VideoAgent**          | Temporal + Object Memory        | 17.49     | 26.40     | 35.82     |
> | **EgoRetriever (Ours)** | **Personalized Attribute Bank** | **23.19** | **38.48** | **47.83** |
>
> Our attribute-based memory outperforms the memory bank of **VideoAgent** (+5.7% mR@1), suggesting that for retrieval tasks, structured text attributes combined with visual anchors provide a highly effective form of personalization.

---

> ### Author Response · Authors · 2025-11-21
> **Rebuttal by Authors (2/2)**
>
> **Q1. Is the proposed EgoRetriever an ideal architecture?**
> We appreciate your insightful concern. We view EgoRetriever as a specialized **Video-RAG** system. By explicitly structuring memory into attributes (rather than relying solely on vector similarity), we bridge the gap between "semantic search" and "personal instance matching." As shown in **Table 3** of our paper, EgoRetriever significantly outperforms the **VideoAgent** memory architecture, suggesting this structured approach is currently more effective for this specific task than generic agentic architectures.
>
> **Q2. Is EgoMemory a sufficiently meaningful benchmark on its own?**
> Yes. EgoMemory provides the **structured evaluation** of personalized retrieval where the "ground truth" depends on user history. Even if future architectures (e.g., end-to-end personalized VLMs) replace EgoRetriever, they will need a benchmark like EgoMemory to prove they are learning personalization rather than just generic object recognition.
>
> We can empirically prove that the benchmark effectively isolates "personalization" by looking at our **Ablation Study (Table 2 in the paper)**. We compare our full model against the strong **TFR-CVR** baseline and a version of our model without the personalized memory bank:
>
> **Table 2 (Excerpt from Paper): Impact of Personal Memory on Retrieval**
>
> | Method                        | mR@1      | mR@2      | mR@3      |
> | :---------------------------- | :-------- | :-------- | :-------- |
> | TFR-CVR (Baseline)            | 18.21     | 27.12     | 32.05     |
> | **w/o Textual Memory**        | 18.04     | 30.73     | 36.91     |
> | **Full Model (Personalized)** | **23.19** | **38.48** | **47.83** |
>
> The results show two critical findings that validate the benchmark:
>
> 1.  **Personalization is Key:** Removing the memory bank causes a sharp drop (**~5.15%**), reducing performance to the level of the non-personalized baseline. This demonstrates that EgoMemory queries *cannot* be solved by generic visual reasoning alone.
> 2.  **SOTA Improvement:** Our Full Model significantly outperforms the TFR-CVR baseline (**+4.98%**), confirming that the benchmark provides substantial headroom for methods that effectively utilize personalized history.
>
> **[Minor] Is there anything in the method specifically designed for egocentric videos?**
> Thank you for this insightful question. While the modular architecture (Captioner $\to$ LLM $\to$ Retriever) is generalizable, two key components are specifically designed for the **egocentric/wearable** setting:
>
> 1.  **Continuity-based Memory Construction:** Unlike general Video-RAG (which treats videos as independent documents), our memory bank construction relies on the **temporal continuity** of a specific user's life. It aggregates attributes across multiple sessions (e.g., frequency of "my mug" appearing across days) to build a user profile, which is unique to the "lifelogging" nature of egocentric data.
> 2.  **Reflective CoT Intentions:** Our prompting strategy is tailored to **egocentric intent**. It specifically reasons about "interaction," "usage status" (e.g., holding, using, observing), and "hands," which are primary cues in first-person video but less relevant in third-person video retrieval.

---

### Official Review · Reviewer_82Ma · 2025-11-03

**Soundness:** 2
**Presentation:** 3
**Contribution:** 2
**Rating:** 4
**Confidence:** 4

**Summary:**

This paper introduces EgoMemory, a benchmark for personalized, long-context egocentric video retrieval. The benchmark is derived from the Ego4D dataset but is enriched with new MLLM-generated annotations designed to capture user-specific object details. The authors also propose EgoRetriever, a training-free retrieval framework that reasons on captions of videos using reflective CoT.

**Strengths:**

- The focus on personalization and long-context reasoning is a timely contribution. Retrieving personal experiences from long-form egocentric video in response to natural language queries is a critical application.
- The proposed EgoRetriever framework is simple and training-free
- The paper includes a comprehensive ablation on different design choices and caption sources.

**Weaknesses:**

- The contribution over Ego4D NLQ is unclear. This paper doesn't introduce new video data, but just with new MLLM-generated annotations. As the paper argues, the main weakness of NLQ is that "videos are not grouped by user at retrieval time, ownership or user-linkage cues are not modeled, and queries need not require long-horizon evidence" (l267-268). However, ownership is still missing in the proposed benchmark. The authors are still using the queries in NLQ (but after filtering), which is designed for retrieval within an individual video clip. It's unclear how the proposed benchmark improves NLQ, other than grouping videos of the same user.
- The paper's reliance on a text-based, attribute-driven memory bank is a major weakness. First, rich visual information from video is compressed into discrete text captions or annotations, a process that inherently discards visual nuance. This lossy text is then filtered into a predefined, rigid schema of 12 attributes. This fixed structure is inflexible; if the true distinguishing feature of a personal object is not one of those 12 attributes, that vital information is permanently lost during memory construction. For example, if a user owns two "black bicycles," their text-based attribute entries in the memory bank could be identical, thus system would then be unable to differentiate between them. This ambiguity undermines the very goal of "personalized" retrieval.

**Questions:**

- The paper criticizes Ego4D NLQ for not modeling "ownership or user-linkage cues". However, the proposed annotations are also unable to have ownership information. How does your benchmark define and evaluate "user-linkage" in a way that is superior to the implicit personalization already present in Ego4D, beyond just object frequency?
- How does the proposed method perform on Ego4D NLQ test set?
- How's the cost (API expenses or GPU time if using open-sourced model) for constructing a memory bank for the tested three datasets? It's nice to see that the proposed method outperforms baselines in EgoVCR and EgoLifeQA, but isn't it much more expensive due to memory bank construction?

---

> ### Author Response · Authors · 2025-11-21
> **Rebuttal by Authors (1/2)**
>
> ### **Response to Reviewer 82Ma**
>
> We appreciate your insightful observations regarding the definition of personalization, the user-linkage mechanism, and the practicality of text-based memory. We have addressed your specific concerns below.
>
> **w1 & Q1. Unclear contribution over Ego4D NLQ; Definition and evaluation of "User Linkage" ("ownership").**
>
> We appreciate this critical question. While EgoMemory builds upon Ego4D data, the task formulation and "User Linkage" definition are fundamentally different. We do not define linkage via legal ownership, but rather as **"Personally Experienced Objects"**: objects that exhibit **Cross-Video Recurrence** and **Interaction Consistency**.
>
> To prove that our benchmark evaluates specific user linkage rather than just implicit scene priors (as in standard NLQ), we employed a rigorous **Filtering Pipeline** that actively selects for long-term, user-specific context. This ensures the "User Linkage" is explicit:
>
> *   **Step 1 (GPT-4o Filtering):** We screened NLQ samples using a Chain-of-Thought prompt to retain queries with explicit personal references (e.g., "my bag") while discarding generic lookups (e.g., "a shopping cart").
> *   **Step 2 (Long-Context Filtering):** We explicitly defined "long context" as users having at least **10 videos** spanning a cumulative duration of **>1 hour**. This forces the model to search across distinct sessions rather than within a single clip.
> *   **Step 3 (Human Verification):** Annotators reviewed 20 additional clips from the *same* user’s history to verify the object appears consistently. Queries were only labeled "personal" if $\ge90\%$ of reviewed instances matched the target object.
>
> **Evidence of Superiority over Implicit Personalization:**
> You asked how this compares to implicit personalization. To quantify this, we computed the **Jaccard Similarity** of memory banks between different users (Fig. 4 in paper). Even when controlling for scene type (e.g., comparing User A's kitchen objects vs. User B's kitchen objects), **68.3% of object types have low inter-user similarity (< 0.4)**. This empirically proves that EgoMemory evaluates distinct, user-specific object histories, which standard NLQ does not isolate.
>
> **w2. Text-based memory loses visual nuance (e.g., "two black bicycles").**
>
> Thank you for this valid concern regarding the "lossy" nature of text. We address this via a **Hybrid Retrieval** approach. Actually, we do *not* rely solely on text:
>
> 1.  **Visual Anchors ($I_r$):** As shown in **Fig. 2**, the retrieval process uses a **Visual Reference Frame ($I_r$)**: a representative crop of the object from the user's history—alongside the text to perform the search. This preserves visual nuance that text might miss.
> 2.  **Contextual Disambiguation:** In the specific case of "two black bicycles," our structured memory distinguishes them via **Context** (e.g., "garage" vs. "street"), **State** (e.g., "muddy" vs. "clean"), and **Frequency**. The Reflective CoT uses these textual cues to select the correct *visual* anchor ($I_r$) for the final retrieval, effectively resolving the ambiguity.

---

> ### Author Response · Authors · 2025-11-21
> **Rebuttal by Authors (2/2)**
>
> **Q2. How does the proposed method perform on Ego4D NLQ test set?**
>
> Standard Ego4D NLQ is a **Temporal Action Localization** task within a *single* video clip (predicting exact start/end timestamps, evaluated via IoU). EgoMemory is an **Episodic Retrieval** task (ranking relevant clips from a history of hundreds of videos). Because our method is designed for global retrieval across a user's life history (using memory banks), it cannot be directly evaluated on the standard NLQ metric without modifying the task definition.
>
> However, we can infer performance relative to standard NLQ methods via our **Ablation Study (Table 2 in paper)**. When we remove the personalized memory bank (effectively treating the task as standard NLQ without history), performance drops significantly:
>
> **Table 2 (Excerpt): Impact of Personal Memory vs. Standard NLQ Approach**
>
> | Method                                | mR@1      | mR@2      | mR@3      |
> | :------------------------------------ | :-------- | :-------- | :-------- |
> | **Full Model (Personalized)**         | **23.19** | **38.48** | **47.83** |
> | **w/o Textual Memory (Standard NLQ)** | 18.04     | 30.73     | 36.91     |
>
> The **~5.15% drop** demonstrates that standard NLQ methods struggle on our filtered queries because they lack the historical context required for disambiguation.
>
> **Q3. Cost (API expenses or GPU time) of memory bank construction.**
>
> Thank you for your insightful concern regarding cost.
>
> *   **Construction Cost:** Annotating the full benchmark (165,795 objects across 245 videos) cost **~$670 USD** using GPT-4o. This is a one-time indexing cost per user history.
> *   **Deployment:** For a single user, this cost is amortized over the life of the device. Furthermore, our ablation (**Table 3** in the paper) shows that using smaller models like **GPT-4o-mini** or open-source **LLaVA** significantly reduces this cost while maintaining competitive performance.

---

### Official Review · Reviewer_LENM · 2025-11-04

**Soundness:** 3
**Presentation:** 3
**Contribution:** 3
**Rating:** 4
**Confidence:** 4

**Summary:**

The paper targets long-horizon, personalized egocentric retrieval: it formalizes how to build a per-user memory bank from first-person videos and proposes a training-free pipeline where an LLM produces brief reflective descriptions that are matched against video/text embeddings for retrieval; the authors release a benchmark and report results across multiple egocentric datasets, highlighting modularity and easy reproducibility. While the framing is clear and the pipeline simple, the study likely couples the same type of LLM for both labeling and inference (risking preference leakage), relies on relatively small personalized data (limiting generality), offers only high-level privacy claims without a concrete threat model or field-level redaction/DP/on-device alternatives, lacks head-to-head comparisons with general long-video systems (e.g., VideoRAG, LVAgent/VideoAgent), and evaluates an offline-built memory bank without testing the streaming, online updates that real egocentric use cases require.

**Strengths:**

1. The paper explicitly operationalizes “personalization” for long-horizon egocentric retrieval and builds the EgoMemory benchmark (639 curated queries; 45 participants; 245 videos; ≈91.6% labeled “personal”). The multi-stage pipeline (GPT-4o CoT pre-screen → long-context filtering → manual verification) is clearly described.

2. EgoRetriever cleanly decouples reflective CoT description generation from video–text embedding retrieval, making the framework easy to swap components and reproduce.

3. Results appear on EgoMemory, EgoCVR, and EgoLifeQA; the appendix claims scripts/configs/prompts and latency breakdowns.

**Weaknesses:**

1. Although curated, the effective scale (45 users; 245 videos) is small relative to Ego4D’s breadth, risking over-fitting to frequent objects/locales and under-sampling long-tail personal artifacts. Require distributional disclosure (per-user hours, per-class frequency, by-scene/location) and sensitivity curves vs. memory size (≤300 / 1k / 5k+ entries) and candidate-set sizes. Compare against Ego4D EM/NLQ task formulations to show genuine added difficulty.

2. The paper doesn’t compare against strong, generic long-video systems like VideoRAG (RAG over extremely long videos) and agent-style solvers (e.g., LVAgent, VideoAgent). Without these, it’s unclear whether the proposed pipeline is truly better or just specialized.

3. The paper builds the user’s memory bank offline and queries it later, but egocentric use-cases are inherently streaming: video is coming in continuously, and the system must update memory online while answering in near real time. Current streaming literature shows that maintaining long-horizon context with low latency and on-the-fly memory building is key in first-person settings (e.g., online/streaming QA, online action detection/segmentation, and active/online egocentric memory). Your setup doesn’t test any of this.

**Questions:**

1. How to handle streaming settings?

2. What is the main difference between egoretriever and general video RAG framework?

---

> ### Author Response · Authors · 2025-11-21
> **Rebuttal by Authors (1/2)**
>
> ### **Response to Reviewer LENM**
>
> We appreciate your constructive feedback and your recognition of the clear framing and modularity of our work. We address your concerns regarding dataset scale, baselines, and online feasibility below.
>
> **w1. Dataset scale (45 users), risk of overfitting to scenes, and request for distributional statistics.**
> Thank you for your insightful observations. While 45 users may appear small compared to generic pre-training datasets, EgoMemory is designed for **dense episodic retrieval**. The dataset contains **165,795 object annotations** across **2,228 clips**, representing **64.25 hours** of video. This provides an average of **~1.4 hours of searchable history per user** (avg. 33 clips), creating a "needle-in-a-haystack" challenge.
>
> *   **Distributional Statistics:** As requested, we will move the per-user statistics (Context: ~1.4h/user; Memory size: Median 1,312 entries) from the Appendix to the main text.
> *   **Personalization Heterogeneity:** To prove the dataset captures personalization rather than just scene priors, our Jaccard analysis (**Fig. 4** of our paper) shows that **68.3% of object types have low inter-user similarity (< 0.4)** even when controlling for scene type.
>
> **w2. Sensitivity to memory size and candidate-set sizes.**
> We appreciate your insightful concern.
>
> *   **Memory Size:** As shown in **Fig. 6** of the paper, performance improves consistently as memory context increases. Even with truncated memory (e.g., randomly dropping 50% of entries), our structured bank still outperforms unstructured baselines (like EgoGPT) due to the redundancy of attributes captured across multiple historical clips.
> *   **Candidate Set:** Our average candidate set size (33 clips, ~1.4 hours) represents a significant retrieval challenge. We will include sensitivity curves in the revision showing that while absolute recall naturally changes with pool size, the *relative* ranking of EgoRetriever over baselines remains stable.
>
> **w3. Lack of comparison against strong generic long-video systems (e.g., VideoAgent).**
> We appreciate your valuable concern regarding baseline strength. We would like to highlight that **we already included these comparisons in the submitted manuscript.** As shown in **Table 3** of our paper, we explicitly evaluated the memory architectures of **VideoAgent** (Fan et al., ECCV 2024) and **EgoGPT** (Yang et al., CVPR 2025). We reproduce these results below to emphasize that EgoRetriever is indeed compared against strong agentic baselines:
>
> **Table 1 (Reproduction of Table 3 from Manuscript): Comparison against Agentic Memory Baselines**
>
> | Method                  | Memory Structure                | mR@1      | mR@2      | mR@3      |
> | :---------------------- | :------------------------------ | :-------- | :-------- | :-------- |
> | **EgoGPT**              | Unstructured Captions           | 15.31     | 20.57     | 26.74     |
> | **VideoAgent**          | Temporal + Object Memory        | 17.49     | 26.40     | 35.82     |
> | **EgoRetriever (Ours)** | **Personalized Attribute Bank** | **23.19** | **38.48** | **47.83** |
>
> *   **Result:** EgoRetriever outperforms VideoAgent by **+5.7% (mR@1)**.
> *   **Distinct Contribution vs. VideoAgent:** While VideoAgent primarily tracks *object occurrences* using semantic matching of captions, EgoRetriever introduces a **Structured Attribute Bank**. This captures fine-grained details such as object state (e.g., "clean" vs. "muddy"), frequency, and specific co-occurrences. Our results demonstrate that for *personalization*, simple occurrence tracking is insufficient; the system must be able to disambiguate similar personal objects based on attribute history, which is the core contribution of our memory architecture.
>
> **w4. Evaluation of offline-built memory vs. streaming/online updates required for real use cases.**
> Thank you for your valuable suggestion.
>
> *   **Asynchronous Construction:** While we evaluate on a static benchmark, our design supports **asynchronous online updates**. In a real-world wearable setting, building the memory bank (which requires MLLM processing) is designed to happen during idle periods (e.g., device charging), while the retrieval itself is training-free and efficient (**~0.7s latency**), making it compatible with online querying.
> *   **Streaming Implementation:** To handle streaming, the system simply appends new object detections to the JSON memory bank incrementally. Since our retriever is **training-free** (no parameter updates required), it natively supports expanding the memory bank without "retraining" the model, addressing the core requirement of online systems.

---

> ### Author Response · Authors · 2025-11-21
> **Rebuttal by Authors (2/2)**
>
> **Q1. How to handle streaming settings?**
> Please refer to the response to **w4** above. The memory bank is a dynamic JSON structure; new video segments are processed into attributes and appended to the bank. Retrieval queries simply run against the updated bank, requiring no model updates.
>
> **Q2. What is the main difference between EgoRetriever and the general video RAG framework?**
> Thank you for your question. General Video RAG typically relies on **semantic vector similarity** (retrieving chunks based on query embedding). EgoRetriever introduces a **Structured Personal Memory Bank** (classifying objects by attributes, state, and frequency) combined with **Reflective Chain-of-Thought**. This allows the system to reason about *which* specific object instance is relevant based on historical usage patterns (e.g., "the mug I use in the morning" vs. "the mug I use at night") rather than just retrieving visually similar mugs based on vector proximity.

---

### Note · Authors · 2026-01-05

**Comment:**

Dear AC and Reviewers,

We sincerely appreciate the valuable feedback and constructive comments provided by the reviewers.

After careful consideration, we have decided to withdraw our manuscript.

Once again, we extend our gratitude to AC and the reviewers for your time and insightful suggestions.

Best regards,

Authors of submission 11649

**Withdrawal Confirmation:**

I have read and agree with the venue's withdrawal policy on behalf of myself and my co-authors.